# Recent Advances (2015–2020) in Drug Discovery for Attenuation of Pulmonary Fibrosis and COPD

**DOI:** 10.3390/molecules28093674

**Published:** 2023-04-24

**Authors:** Atukuri Dorababu, Manikantha Maraswami

**Affiliations:** 1Department of Chemistry, SRMPP Government First Grade College, Huvinahadagali 583219, India; 2Department of Chemistry, Abzena LLC., 360 George Patterson Blvd, Bristol, PA 19007, USA

**Keywords:** pulmonary fibrosis, COPD, TGF-β1, Smad signaling pathway, collagen deposition

## Abstract

A condition of scarring of lung tissue due to a wide range of causes (such as environmental pollution, cigarette smoking (CS), lung diseases, some medications, etc.) has been reported as pulmonary fibrosis (PF). This has become a serious problem all over the world due to the lack of efficient drugs for treatment or cure. To date, no drug has been designed that could inhibit fibrosis. However, few medications have been reported to reduce the rate of fibrosis. Meanwhile, ongoing research indicates pulmonary fibrosis can be treated in its initial stages when symptoms are mild. Here, an attempt is made to summarize the recent studies on the effects of various chemical drugs that attenuate PF and increase patients’ quality of life. The review is classified based on the nature of the drug molecules, e.g., natural/biomolecule-based, synthetic-molecule-based PF inhibitors, etc. Here, the mechanisms through which the drug molecules attenuate PF are discussed. It is shown that inhibitory molecules can significantly decrease the TGF-β1, profibrotic factors, proteins responsible for inflammation, pro-fibrogenic cytokines, etc., thereby ameliorating the progress of PF. This review may be useful in designing better drugs that could reduce the fibrosis process drastically or even cure the disease to some extent.

## 1. Introduction

Pulmonary fibrosis (PF) is defined as scarring of the lungs over time, and the symptoms include dry cough, shortness of breath, feeling tired, weight loss, and nail clubbing. In addition, complications such as pulmonary hypertension, respiratory failure, pneumothorax, and lung cancer may also be observed. PF comprises the gradual exchange of normal lung parenchyma with fibrotic tissue, which may subsequently lead to irreversible decreases in oxygen diffusion capacity and stiffness. Reported causes of PF include environmental pollution [1], some types of medications (such as amiodarone, bleomycin, busulfan, methotrexate, nitrofurantoin [2], etc.), interstitial lung diseases, and SARS infection. Conditions such as CS, some connective tissue diseases, tuberculosis, etc., may cause PF as a secondary effect. In some cases, PF is diagnosed without any cause; such cases are termed “idiopathic PF”. In a few cases, genetic predisposition is evident in some patients, wherein a mutation in surfactant protein C has been reported in families with a history of PF. Alongside this, in about 15% of PF patients, autosomal dominant mutations in the telomerase RNA component (*TERC*) or *TERT* genes encoding telomerase have been identified.

Primarily, the lung tissues in patients with PF are degenerated, wherein the alveolar units are damaged. With excess collagen accumulation, the mass of the lungs increases, thereby reducing the elasticity of the lungs and, in turn, decreasing their oxygen-holding capacity (Figure 1). The literature indicates that differentiation of fibroblasts into myofibroblasts is a key step in the fibrotic process. It has been reported that follistatin-like 1 factor promotes PF by enhancing lung fibroblast differentiation, proliferation, migration, and invasion through p38 and c-Jun N-terminal kinase (JNK) signaling [3]. There are a variety of models available to study PF, of which the bleomycin-induction model is the most commonly used [4]. Although there is no cure or permanent treatment for IPF, a few treatment options are available. Some types of medications are prescribed to slow down the processes that lead to fibrosis. For mild IPF, the pharmacological agent pirfenidone prevents scarring. Additionally, the anti-fibrotic agent nintedanib functions through the inhibition of tyrosine kinase receptors [5]. A clinical trial report revealed that nintedanib reduced lung-function decline and acute exacerbations [6]. Approximately five million people have been affected by PF worldwide. Apart from medications, lung transplantation is also an option in severe cases.

In the recent literature, D.S. Glass et al. described molecular mechanisms and possible treatment approaches of IPF [7]. Traditional Chinese medicine has been demonstrated as an efficient treatment for PF [8]. In addition, inhibition of MMPs has been considered as a potent therapeutic treatment for IPF [9]. L. Kolilekas et al. highlighted existing and emerging treatment options for IPF treatment, including phase II–IV trials [10]. Formulation of evidence-based clinical practice was studied extensively and discussed by S. Homma et al. [11]. In this review, various medications reported recently are critically evaluated. In addition, the review is classified into sections, e.g., biomolecule-based and synthetic drugs for PF/IPF and COPD.

### 1.1. Idiopathic Pulmonary Fibrosis (IPF)

Amongst the fibrotic lung diseases, IPF is the most common, with limited treatment options. The majority of idiopathic pulmonary fibrosis patients are male smokers of more than 60 years of age [12]. Drug treatment can only improve the lung function of pulmonary fibrosis patients with mild-to-moderate symptoms. However, drug therapy does not exert substantial effects on patients’ lifespan [13]. Hypoxia—a characteristic of PF—induced hypoxia-inducible factor 1-alpha (HIF-1α), which contains an oxygen-regulated α-subunit [14].

### 1.2. Attenuation of PF by Biomolecule-Based Inhibitors

The literature indicates that scutellarin has inhibitory properties against liver fibrosis and myocardial fibrosis [15,16]. Based on this fact, K. Miao et al. demonstrated the effect of scutellarin **1** (Table 1, entry 1) on PF [17]. Scutellarin treatment mitigated lung injury and fibrosis, and the toxicity test revealed that scutellarin is non-toxic based on the levels of aspartate aminotransferase (AST), urea, Cr, lactose dehydrogenase (LDH), and creatine kinase (CK). Interestingly, lowering of alanine transaminase (ALT) levels infers a protective effect of scutellarin on the liver. Furthermore, scutellarin reduced the expression levels of fibrosis-specific indicators such as collagen I and fibronectin. To investigate the inhibition of fibroblast differentiation, transforming growth factor beta-1 (TGF-β1)-administered groups were treated with scutellarin, where a dramatic decline in the expression of α-smooth muscle actin (α-SMA), collagen I, and fibronectin was noted, and the apparent therapeutic effect was observed at a 50 µM concentration. Alongside this, the mechanism hypothesized in hindering the fibroblast differentiation is the inhibition of TGF-β/Smad signaling, as scutellarin markedly inhibited the p-Smad2 and p-Smad3 expression levels in a dose-dependent manner. Furthermore, scutellarin exerted remarkable reductions in p-P85-α and phosphorylated protein kinase-B (p-Atk) expression levels, indicative of diminished fibroblast proliferation. Additionally, at the concentration of 50 µM, scutellarin was found to facilitate fibroblast apoptosis by regulating the B-cell leukemia/lymphoma protein (Bcl-2)/Bax pathway.

EM703 **2** (Table 1, entry 2)—a 12-membered ring macrolide derivative of erythromycin—has been reported to suppress the activation of nuclear factor (NF)-κB and the production of interleukin-8 [18]. Y. J. Li et al. demonstrated anti-inflammatory and anti-fibrotic properties of EM703 [19]. EM703 treatment significantly reduced the hydroxyproline content in lung tissue compared to bleomycin-treated groups, indicating a potent anti-fibrotic effect of EM703. Additionally, TGF-β-induced MLg2908 (mouse lung fibroblast line) proliferation was reported to be diminished. Production of soluble collagen by MLg2908 was significantly increased by TGF-β, while EM703 treatment remarkably inhibited the concentration of soluble collagen. Alongside this, the reduced expression of Smad3 and Smad4 mRNA in MLg2908 cells caused by the addition of TGF-β was reversed to control levels with EM703 treatment. Moreover, the expression of Smad3/4 and p-Smad2/3 proteins in MLg2908 was found to be abated by pretreatment with EM703, whereas the increased expression of p-Smad2/3 protein caused by TGF-β exposure for 12 h could not be controlled by EM703 treatment. These findings indicate that the mechanism involved in the inhibition of PF is the inhibition of TGF-β signaling, which mediates fibroblast proliferation and extracellular matrix production.

Salidroside **3** (Table 1, entry 3) is a phenolic glycoside isolated from the herb *Rhodiola rosea*. Literature indicates that salidroside can abate chronic hypoxia-induced pulmonary arterial hypertension in mice [20], in addition to a suppressive effect on TGF-β1 [21]. In view of this, H. Tang et al. explored salidroside’s effects on fibrotic injuries to the lungs [22]. Salidroside exhibited a dose-dependent inhibitory effect on the upregulation of hydroxyproline, indicating preventive activity against fibrotic lesions. Alongside this, the suppressive effect on detrimental inflammatory infiltration by salidroside (200 mg/kg) demonstrated anti-inflammatory and antioxidative properties that work via the NF-κB and nuclear factor erythroid 2 (Nrf-2) signaling pathways. Through blocking the expression levels of bleomycin-induced TGF-β1 and hindering the phosphorylation of Smad2/3, salidroside (50, 100 mg/kg) inhibited bleomycin-induced TGF-β/Smad signaling transduction in lung tissues. Furthermore, reduction in bleomycin-induced E-cadherin suppression was achieved by salidroside at a dose of 200 mg/kg. Moreover, bleomycin-upregulated protein expression of epithelial–mesenchymal transition (EMT) makers, fibronectin, and α-SMA was downregulated by salidroside (dose-dependent effect, maximum at 200 mg/kg), indicating that the drug has an inhibitory effect on bleomycin-induced mesenchymal shifts in rat lung tissues.

Ajulemic acid **4** (Table 1, entry 4)—a synthetic analog of tetrahydrocannabinol—has been reported to exert potent anti-fibrotic effects in experimental models of systemic sclerosis [23,24]. Hence, M. Lucatelli et al. attempted to investigate the anti-fibrotic effect of ajulemic acid on human PF [25]. Ajulemic acid administration markedly diminished the number of inflammatory cells at day 8 and attenuated the collagen deposition at day 14 in bleomycin-treated mice compared to mice treated with bleomycin alone. The decline in collagen accumulation was also proven by the hydroxyproline-reducing effect of administering ajulemic acid (1 mg/kg/day and 5 mg/kg/day) on day 21. At similar concentrations, ajulemic acid abated the effects of TGF-β1. Meanwhile, connective tissue growth factor (CTGF) levels were markedly improved in tissue samples treated with ajulemic acid. Blockage of myofibroblast differentiation was evident based on the significant reduction in the number of α-SMA-positive cells at day 14. The authors presented an interesting investigation, wherein the effects of ajulemic in delayed treatment at the fibrogenic stage were discussed. After day 8 of bleomycin administration at concentrations of 1 and 5 mg/kg/day, the collagen deposition was reduced, and samples exhibited low hydroxyproline content. Meanwhile, the other factors that were increased/decreased by bleomycin treatment were balanced by delayed ajulemic acid treatment. These findings suggest that both the therapeutic ajulemic acid treatment and preventive treatment were equally important.

The non-structural protein of the hepatitis C virus, NS5ATP9, has been reported to attenuate liver fibrosis through downregulation of the TGF-β1/Smad3 signaling pathway, suppressing cell proliferation and promoting apoptosis [26]. Similarly, the literature shows that the reverse transcriptase inhibitor tenofovir alafenamide fumarate (TAF) **5** (Table 1, entry 5) suppressed fibrosis in patients with hepatitis B virus (HBV) infection [27]. In view of these revelations, Li et al. investigated the anti-fibrotic effect of TAF in PF and the relationship between TAF and NS5ATP9 in human lung fibroblasts in vitro [28]. In this study, silencing of the protein NS5ATP9 resulted in upregulation of the expression levels of α-SMA and collagen 3α1, while downregulation of these protein levels was noted with overexpression of NS5ATP9. Moreover, positive results were obtained, wherein TAF was reported to upregulate the expression of NS5ATP9 at the RNA level in bleomycin-induced mice on days 28 and 35. Administration of TAF (5.125 mg/kg) for three weeks showed slightly improved body weight loss, and the mortality rate was found to decrease from 40–50% to 0–20% on days 21, 28, and 35. Alongside this, TAF attenuated histopathological changes in bleomycin-induced PF, and the changes were intermediate between the control group and the bleomycin group. Compared to the bleomycin group, significant reductions in the expression of TGF-β1, collagen 3α1, α-SMA, and fibronectin were evident as a result of the TFA treatment on day 21, and these results were consistent with those found on days 28 and 35. Furthermore, TAF at concentrations of 250 µM/L and 500 µM/L was found to inhibit phosphorylation of Smad3 in lung fibroblasts, indicating a remarkable regulatory effect on the TGF-β1/Smad3 signaling pathway.

Considering the protective effects of tannic acid **6** (Table 1, entry 6)—a natural polyphenol obtained from tarapods, gallnuts, and leaves of certain plants [29]—against cardiac hypertrophy/fibrosis and liver fibrosis in mice [30], E. B. Reed et al. studied the anti-fibrotic properties of tannic acid in a murine model of PF [31]. Tannic acid treatment had dramatic inhibitory (dose-dependent effect, maximum activity at 3 µM) activity against TGF-β-induced expression of collagen-1 and α-SMA, indicating that tannic acid is a potent inhibitor of myofibroblast differentiation. Moreover, the inhibition of myofibroblast differentiation was evident from the significant inhibition of sustained Smad2 phosphorylation induced by TGF-β over a 48 h duration. Additionally, reduced collagen deposition and low hydroxyproline contents indicated inhibition of PF by tannic acid.

Alongside the broad-spectrum pharmacology of emodin **7** (Table 1, entry 7) extracted from *Rheum palmatum* L., moderate suppressive effects against pancreatic and hepatic fibrosis have also been reported for emodin [32]. The study of the anti-fibrotic effects of emodin was continued by S. Tian et al., where the authors explored emodin for the attenuation of bleomycin-induced PF via anti-inflammatory and antioxidant activities [33]. The administration of emodin (20 mg/kg for three consecutive weeks) to bleomycin-treated rats moderated the PF via reducing hydroxyproline levels. Alongside this, emodin demonstrated anti-inflammatory effects in bleomycin-treated rats, wherein the obstruction of excessive production of pro-inflammatory cytokines such as interleukin (IL)-1β, IL-6, and TNF-α was reported. An approximately twofold reduction in pro-inflammatory cytokine production was observed. The enhancement of antioxidant levels (greater than twofold) of glutathione (GSH), superoxide dismutase (SOD), and GSH-Px compared to controls, along with the additional accumulation of Nrf2 and increase in heme oxygenase (HO)-1 levels, indicated antioxidant effects of emodin via the Nrf2 signaling pathway. Furthermore, emodin attenuated the phosphorylation of Smad2/3 and suppressed the expression of TGF-β1. Furthermore, emodin hindered the decreased expression of E-cadherin. Compared to the TGF-β1 group, more than twofold suppression of protein and mRNA expression of vimentin, fibronectin, and α-SMA was evident with the emodin treatment, indicating the therapeutic efficacy of emodin against PF.

Oridonin **8** (Table 1, entry 8)—a diterpenoid extracted from *Rabdosia rubescens*—has been reported by many studies for its potent activity against various lung diseases [34,35,36]. In view of this, Y. Fu et al. explored oridonin for the inhibition of myofibroblast differentiation [37]. Initial cell viability tests of oridonin on Medical Research Council (MRC)-5 cells indicated that oridonin has toxic effects up to 10 µM concentration. At a concentration of 10 µM, oridonin showed a gradual reduction in TGF-β1-induced mRNA and protein of α-SMA and collagen I expression; the effects were more pronounced compared to pirfenidone (2 mM). Furthermore, with respect to morphological changes, oridonin treatment apparently decreased the structural destruction, inflammation, and collagen deposition in mouse lungs compared to the control group. Additionally, reduced pathology scores were noticed, which almost approached the control values. However, only a slight reduction in the pulmonary index was observed compared to the model, while pirfenidone (300 mg/kg) exhibited a comparatively higher pulmonary index. Alongside this, oridonin blocked bleomycin’s upregulation of α-SMA and collagen I in the lung tissues of mice. Furthermore, the therapeutic efficacy of oridonin was confirmed by reduced phosphorylation of Smad2/3, thereby inhibiting the corresponding protein expression and blocking Smad/TGF-β signaling transduction in lung tissues.

Necrostatin-1 **9** (Table 1, entry 9) is reported to reduce lipopolysaccharide-induced lung injury and inflammation [38,39]. However, the role of necrostatin-1 in the attenuation of PF is not clear. F. Mou et al. demonstrated the efficacy of necrostatin-1 against PF [40]. In this study, elevated protein expression and mRNA expression of RIPK1 and RIPK3 induced by bleomycin were reduced by necrostatin-1 administration. This was also supported by remarkable downregulation of TGF-β1-induced receptor-interacting serine/threonine-protein kinase 1 (RIPK1) and RIPK3 expression in MRC-5 cells. The elevated inflammatory cell infiltration, collagen deposition, and pulmonary interstitial congestion wee significantly reduced by necrostatin-1 treatment compared to mice treated with bleomycin alone. Meanwhile, there were slight reductions in the expression of α-SMA, collagens I and IV, fibronectin, and TGF-β, indicating decreased extracellular matrix (ECM) expression. Furthermore, the protein expression of these factors induced by TGF-β1 in MRC-5 cells was also suppressed by necrostatin-1 administration. However, the expression levels were comparatively higher than those of the control group. Finally, necrostatin-1 could inhibit the proliferation of pulmonary TGF-β1-induced fibroblasts.

Ginsan is a polysaccharide obtained from the roots of *Panax ginseng* by C. A. Meyer. It has been demonstrated to correct the imbalance of cytokines related to fibrosis. In their study, J-.Y. Ahn et al. investigated the inhibitory effect of ginsan on TGF-β-mediated fibrotic processes [41]. In the inhibitory study of TGF-β-induced accumulation of ECM components, at a concentration of 12.5–25 µg/mL, ginsan had an effect on α-SMA protein levels, while marked inhibitory activity was observed at the concentration of 50–200 µg/mL. Meanwhile, the accumulation of other ECM components such as collagen I, fibronectin, and procollagen type I induced by TGF-β in National Institutes of Health (NIH)/3T3 cells was also suppressed. In addition, ginsan also reduced the α-SMA and fibronectin (FN) levels in normal human lung fibroblasts, indicating that the activity of ginsan is non-specific. Moreover, ginsan pretreatment resulted in significantly decreased phosphorylation of Smad2 and Smad3. TGF-β-promoted CAGA luciferase activity was decreased upon the administration of ginsan, whether before or after treatment. Furthermore, ginsan reduced the extent of phosphorylation of extracellular signal-regulated kinase (ERK) and Akt induced by TGF-β. However, JNK and dp38 were not phosphorylated by ginsan. This implies downregulatory activity of ginsan that is both Smad-dependent and Smad-independent. Meanwhile, elevated expression levels of TGF-β receptors such as TβRI and TβRII were significantly decreased by the addition of ginsan. Furthermore, in the in vivo studies, the administration of ginsan at a concentration of 2 mg/kg on alternate days for two weeks led to noteworthy suppression of collagen and α-SMA expression levels compared to the group treated with bleomycin alone.

B. Li et al. attempted to explore ouabain **10** (Table 1, entry 10) for inhibitory activity against bleomycin-induced PF [42], considering that ouabain can inhibit lung fibroblast activation and epithelial–mesenchymal plasticity via obstructing the TGF-β-Smad signal pathway [43,44]. Initially, the administration of ouabain (0.5 mg/kg) led to a reduction in the body weight loss (*p* < 0.05) from day 7 to day 14 in bleomycin-induced mice. Moreover, attenuation of lung injury and decreased collagen deposition were observed with ouabain treatment, and the effects were consistent with the Ashcroft scoring (*p* < 0.05). Alongside this, bleomycin-promoted elevated expression levels of fibrosis markers such as collagen I, fibronectin, and α-SMA were significantly attenuated by ouabain treatment. Additionally, the mRNA levels of collagen I, fibronectin, and α-SMA were found to be abated in lung tissues. At a concentration of 100 nM, ouabain markedly inhibited the fibroblast proliferation in MRC-5 cells when compared to the control group at 24 h, 48 h, and 72 h. Furthermore, apoptosis of lung fibroblasts was noted after ouabain treatment (100 nM) for 48 h, as evident from the significantly enhanced early-apoptotic and late-apoptotic cell populations as well as necrotic cell populations compared to the control group.

Accumulating evidence indicates that insulin-like growth factor-1 (IGF-1) promotes the proliferation, migration, and collagen production of fibroblasts [45], and that the inhibition of the IGF-1 pathway would alleviate bleomycin-induced lung injury in animal models [46]. Furthermore, the antidiabetic agent metformin **11** (Table 1, entry 11) has been demonstrated as an IGF-1 inhibitor [47]. This fact was the basis for the investigation of metformin for its anti-fibrotic properties in cases of PF by H. Xiao et al. [48]. The administration of metformin significantly increased the survival rate in the bleomycin-treated mice, where the median survival time increased from 11 days to 20.5 days. Alongside this, the elevated levels of collagen fiber deposition and other pathological changes were reduced in metformin-treated mice compared to the bleomycin-treated group. Additionally, Masson trichrome staining and immunohistochemistry (IHC) staining indicated that metformin administration decreased the deposition of collagen fibers. The Western blotting test revealed improvements in PF, as evident by the reduced contents of hydroxyproline and fibronectin, which are specific markers of PF. The body weight loss of bleomycin-treated mice was remarkably decreased with metformin treatment, and a similar effect was observed with combined treatment using metformin and pirfenidone. Moreover, metformin treatment markedly suppressed the increased expression levels of IGF-1 and its downstream cytokine phosphoinositide 3-kinase (PI3K). Furthermore, the process of fibroblasts’ differentiation into myofibroblasts was alleviated through the reduction in α-SMA levels in bleomycin-treated mice. In this study, the efficacy of metformin in attenuating PF was found to be similar to that of pirfenidone or the combined effect of both drugs.

S.A.M. Saghir et al. investigated the improved anti-fibrotic effect of thymoquinone **12** (Table 1, entry 12) in the form of its nanoparticles in rats compared to the plain drug [49]. The prepared nanoparticles (TQ-NPs), around 20 nm in size, possessed encapsulation efficiency of 80%. The severity of gross pneumonia in bleomycin-induced rats was reduced with subsequent treatment with TQ-NPs. In the case of pulmonary TGF-β1 and IL-10, bleomycin induction elevated the levels of TGF-β1 (56.38 pg/mL) and IL-10 (408.9 pg/mL) compared to controls, while the administration of TQ-NPs significantly reduced the levels of TGF-β1 (22.19 pg/mL) and IL-10 (204.5 pg/mL) to levels comparable with controls, indicating attenuation of fibrosis. Histopathological findings revealed that bleomycin treatment led to obliteration of lung alveoli because of severe infiltration of fibroblasts, alveolar macrophages, and deposition of collagen in the alveolar septa. These changes were gradually reduced by TQ-NPs treatment. Alongside this, bleomycin injection markedly increased the inducible nitric oxide synthase (iNOS) (iNOS) immunoreactivity within the peribronchial and interalveolar regions. However, co-treatment with TQ-NPs showed a reduction in iNOS immunoexpression within the thickened peribronchial and interalveolar regions. Furthermore, bleomycin-treated rats exhibited intense types of collagen fibers, while the rats co-treated with TQ-NPs showed remarkable collagenolysis. These facts indicate potent anti-fibrotic effects of TQ-NPs.

H. Chen et al. investigated the effects of curcumin **13** (Table 1, entry 13) on the arterial blood gas index of rats with PF induced by paraquat [50]. Initial comparisons of arterial blood pH for days 1 and 5 indicated no significant difference. The partial pressure of oxygen (PaO_2_) in the arterial blood of the curcumin group was significantly higher compared with the paraquat group, while the partial pressure of carbon dioxide (PaCO_2_) was lower than that of the paraquat group, revealing that the curcumin could regulate the gas index. Curcumin treatment reduced the Smad4 and Smurf levels as compared with the paraquat group, thereby causing dysfunction of the TGF-β1/Smad signaling pathway and, subsequently, inhibiting the development of PF. Expression of IL-8 can induce fibroblast proliferation and lead to collagen accumulation, whereas interferon gamma (IFN-γ) inhibits the fibroblast proliferation and promotes collagen synthesis. It has been reported that curcumin can regulate the levels of IL-8 and IFN-γ in the body. In this way, curcumin has inhibitory effects on paraquat-poisoned PF.

Polydatin **14** (Table 1, entry 14)—an isolate of the Chinese medicinal herb Polygonum *Cuspidatum Sieb.* Et *Zucc*—has been used in the treatment of lung injuries and to alleviate ROS and bleomycin-induced EMT and PF [51]. Meanwhile, Y. Liu et al. conducted a comparative study on resveratrol and polydatin with respect to the inhibition of PF [52]. The A549 cell viability of polydatin (10–120 µM) was reported to be higher than 90%. The effect of polydatin (30 and 90 µM) on the EMT of A549 cells indicated that polydatin significantly inhibited the reduction in E-cadherin and the increase in Col I and α-SMA levels induced by TGF-β1 (10 ng/mL). Furthermore, gene expression for these fibrosis markers declined. These effects were found to be dose-dependent. In the pathological study, both the resveratrol and polydatin at the same dose demonstrated similar effects. However, at higher concentrations, polydatin exhibited a stronger protective effect. Moreover, the resveratrol and polydatin markedly downregulated the expression of hydroxyproline, subsequently reducing collagen deposition. Combined administration of resveratrol and polydatin (40 and 160 mg/kg) attenuated the elevated levels of pro-inflammatory factors such as TNF-α, IL-6, and IL-13. Alongside this, the elevated levels of MDA and MPO and the decreased T-SOD levels were restored by the resveratrol and polydatin treatment at concentrations of 40 and 160 mg/kg, respectively, whereas polydatin exerted these effects to a certain extent at concentrations as low as 10 mg/kg. Additionally, polydatin rendered promising effects on signaling pathways, where the increased expression levels of TGF-β1 and the phosphorylation of its downstream signals (Smad2, Smad3, and ERK1/2) were inhibited by the polydatin administration.

The polysaccharides isolated from the Chinese herb *Astragalus* (APS) demonstrated antitumor and anti-fibrotic activity [53]. To investigate the role of APS in PF, R. Zhang et al. studied the inhibition of the epithelial–mesenchymal transition and NF-κB pathway activation [54]. APS reduced bleomycin-induced collagen deposition and hydroxyproline content. In addition, APS attenuated the upregulation of collagen-1a1 and fibronectin at the mRNA level. Western blotting results indicated a decline in vimentin expression and an increase in E-cadherin expression. Furthermore, APS decreased TGF-β1-induced PF through inhibition of the NF-κB pathway. 

Paeoniflorin **15** (Table 1, entry 15)—a main constituent of the Chinese medicinal herb *Paeonia lactiflora*—has been reported to inhibit the expressions of intracellular adhesion molecule (ICAM-1), monocyte chemoattractant protein-1 (MCP-1), IL-6, and TNF-α in endothelial cells. In view of this, Y. Ji et al. investigated the effects of paeoniflorin on bleomycin-induced PF in mice [55]. The decreased survival rate (50%) in bleomycin-treated mice was gradually increased (87.5%) by the paeoniflorin treatment (50 and 100 mg/kg). At a concentration of 100 mg/kg, changes in histopathology were significantly alleviated. At similar concentrations of paeoniflorin, the production of ECM was attenuated in lung tissues, and the effect was equivalent to that of prednisone (6 mg/kg). Meanwhile, the elevated hydroxyproline content was significantly reduced to 43.7% with paeoniflorin administration. Alongside this, levels of α-SMA were markedly inhibited (72.6%) by paeoniflorin (100 mg/kg) compared to prednisone (91.8%) at a concentration of 6 mg/kg. Furthermore, paeoniflorin (100 mg/kg) suppressed the elevated phosphorylation of Smad2/3 and Smad4 expression. Conversely, upregulation of Smad7 was observed after paeoniflorin administration. The anti-fibrotic cytokine IFN-γ was reported to be decreased with the addition of bleomycin, and the effect was reversed with paeoniflorin administration, while paeoniflorin demonstrated only slight inhibition of MMP-1 and TIMP-1, which are responsible for the degradation of type I collagen in PF. The abovementioned facts indicate remarkable effects of paeoniflorin against PF.

Mangiferin **16** (Table 1, entry 16)—an isolate of peels and kernels of mango fruits and bark—has been investigated for the treatment of bleomycin-induced PF through inhibiting toll-like receptor protein (TLR4)/p65 and the TGF-β1/Smad2/3 pathway [56]. The pulmonary index of bleomycin-induced mice was significantly reduced by mangiferin administration (40 mg/kg) for 14 days, while mangiferin improved the decreased body weight in bleomycin-treated mice. The mangiferin instillation demonstrated a remarkable attenuation of significantly increased infiltration of inflammatory cells (i.e., neutrophils, lymphocytes, and macrophages) infiltration. Moreover, the hydroxyproline content—a major component in collagen—was found to be fourfold higher compared to controls (1.2 µg/mg), while mangiferin administration decreased the hydroxyproline content compared with the bleomycin-induced group. The representative protein α-SMA in the mesenchyme was alleviated (*p* < 0.01) by mangiferin treatment. The study concerning the investigation of mechanisms of action revealed suppression of P65 phosphorylation; this, in turn, indicates that TLR4 was the target of mangiferin that inhibited the commencement of inflammation in mice. Furthermore, mangiferin has been reported to restore normal epithelial morphology and expression of the marker proteins E-cadherin and α-SMA, indicating the suppression of the EMT process. Alongside this, the inhibition pathway of mangiferin was shown to be the TGF-β1/Smad pathway.

Based on evidence of the anti-fibrotic properties of hydroxysafflor yellow A (HSYA) **17** (Table 1, entry 17)—an active component of *Carthamus tinctorius* L.—M. Jin et al., investigated the inhibitory effect of HSYA against PF in mice [57]. HSYA at concentrations of 40 mg/kg and 60 mg/kg recovered the decreased body weights in bleomycin-induced mice. In particular, the concentration of 60 mg/kg demonstrated an excellent effect (from day 3 to day 21). Regarding the gas index, the reduced blood pH and PaO_2_ and the increased PaCO_2_ were restored to normal levels by HSYA administration (60 mg/kg). The histopathological changes in murine livers were slightly ameliorated by HSYA treatment, and the effects were found to be dose-dependent. Additionally, the hydroxyproline contents in the lungs were reduced (*p* < 0.05) in the high-dose HSYA groups. HSYA instillation resulted in decreased numbers of α-SMA-positive cells, and the areas of focal consolidation in the lungs were smaller compared to those of mice treated with bleomycin alone. Morphological aberrations in A549 cells induced by TGF-β1 were restored by HSYA treatment (high dose). Moreover, HSYA decreased the collagen I mRNA levels in A549 cells and enhanced the expression of E-cadherin mRNA. Alongside this, attenuation of Smad phosphorylation in A549 cells was evident upon treatment with HSYA. The same research group continued their investigation of the effects of HSYA against PF but at a higher drug dose (80 mg/kg) [58]. It was reported that HSYA at the higher dose demonstrated greater anti-fibrotic effects.

X. Song et al. aimed to investigate the inhibitory activity of all-*trans* retinoic acid (ATRA) **18** (Table 1, entry 18) against bleomycin-induced lung fibrosis through downregulation of the TGF-β1/Smad3 signaling pathway in rats [59]. The ATRA-treated groups showed thinner alveolar walls and lower degrees of edema and infiltration of macrophages, neutrophils, and lymphocytes, and fibroblasts, and erythrocytes. Transmission electron microscopy (TEM) results revealed that the ARTA treatment decreased collagen I expression and the number of collagen fibers. Additionally, ARTA administration restored bleomycin-induced changes in α-SMA and E-cadherin. ARTA significantly decreased the mRNA expression of TGF-β1, Smad3, p-Smad3, and zinc finger E-box-binding homeobox 1 (ZEB1), ZEB2, and high-mobility group AT-hook 2 (HMGA2), regulated by the TGF-β1/Smad3 signaling pathway.

Reports indicate that pycnogenol (PYC) extracted from pine bark (pine bark extract) has anti-inflammatory and preventive effects against wrinkle formation through TGF and type I procollagen [60]. In view of this, J. Ko et al. studied the inhibitory activity of PYC against cigarette smoking (CS)-induced PF [61]. PYC-treated (10 mg/kg) mice showed significantly decreased inflammatory cell counts compared to CS- and liposaccharide (LPS)-induced mice, and the effects were dose-dependent. In terms of pro-inflammatory cytokines, a remarkable decline in IL-6, IL-1β, and TNF-α levels was observed with PYC treatment. Alongside this, PYC attenuated extensive collagen deposition as compared with CS- and LPS-exposed mice. PYC suppressed the expression levels of TGF-β1 and collagen. The anti-fibrotic effects demonstrated by PYC at a concentration of 20 mg/kg were almost comparable with those of roflumilast (10 mg/kg).

J. Li et al. investigated the effects of andrographolide (AG) **19** (Table 1, entry 19) on the proliferation and myofibroblast differentiation of fibroblasts in vivo and in vitro [62], considering its anti-inflammatory properties and suppression of oxidative stress and EMT in the lungs [63]. At a concentration of 10 mg/kg, AG administration reduced the number of myofibroblasts expressing α-SMA compared to mice treated with bleomycin alone. In addition, the numbers of fibroblast-specific protein-1 (FSP-1)/α-SMA-positive cells were also significantly decreased by AG treatment, indicating suppression of fibroblast proliferation and differentiation. Alongside this, AG treatment reduced TGF-β1-induced cell proliferation protein and proliferating cell nuclear antigen (PCNA). The FACS analysis demonstrated that the AG administration increased cell apoptosis in the basal and TGF-β1-stimulated NIH 3T3 fibroblasts, and the mechanism of apoptosis was reported to be the inhibition of the caspase pathway. The elevated levels of fibronectin and collagens I and II in NIH 3T3 cells and PLFs induced by TGF-β1 were significantly attenuated by AG instillation. AG has been reported to have inhibitory effects on the TGF-β1-induced Smad2/3 signaling pathway in NIH 3T3 fibroblasts and pulmonary lung fibroblasts (PLFs). Meanwhile, significant suppression of Erk-1/2 phosphorylation was observed with AG treatment as compared with cells treated with TGF-β1 alone, indicating the remarkable anti-fibrotic effects of AG.

Nagilactone D (NLD) **20** (Table 1, entry 20) is a dinorditerpenoid isolated from *Podocarpus nagi* and known to be a potent inhibitor of TGF-β1-induced elongated fibroblast-like morphologies [64]. A. Li et al. aimed to investigate the effects of NLD against bleomycin-induced PF [65]. NLD co-treatment (1 and 2 µM) inhibited the upregulation of TGF-β1-induced expression of fibrotic markers such as collagens I and II, fibronectin, α-SMA, and CTGF. At similar concentrations, NLD inhibited TGF-β1-induced upregulation of TβR I in WI-38 VA-14 and HFL cells, whereas no effect was observed on the protein levels of TβR II in both cell types. TGF-β1-stimulated phosphorylation of Smad2 was markedly inhibited with NLD treatment (2 µM), in addition to suppression of the nuclear translocation of Smad2. Meanwhile, co-treatment with NLD resulted in a marked decrease in SBE-luciferase activity. Furthermore, NLD administration (1 or 3 mg/kg/day) for 14 or 21 consecutive days significantly decreased the bleomycin-induced inflammatory responses in the alveolar interstitium and bronchial walls. On the other hand, NLD reduced the expression of fibrotic markers and corresponding protein levels. These results indicate that NLD could be investigated further to test its efficacy.

A biologically active molecule, baicalin **21** (Table 1, entry 21), obtained from *Scutellaria baicalensis* Georgi, has been found to exhibit anti-apoptotic [66] and anti-inflammatory properties. Considering these facts, H. Zhao et al. explored baicalin for its anti-fibrotic effects on bleomycin-induced PF [67]. Baicalin (50 mg/kg) markedly reduced bleomycin-induced pathological changes, including collagen deposition and damage to the lung structure and interstitium. Additionally, the bleomycin-induced expression of collagens I and II was suppressed by baicalin treatment. In terms of inflammatory infiltration, the total cell numbers were decreased (4.76) by baicalin administration as compared with mice treated with bleomycin alone (5.26). Moreover, ELISA quantification revealed that baicalin decreased the expression of TNF-α (364) and TGF-β1 (51.2) as compared with bleomycin-treated mice (394 and 65.92, respectively). Bleomycin induction reduced the activity of GSH-px (738), T-SOD (186), and GSH (795), which are associated with oxidative stress and were reported to be enhanced by baicalin to 769, 212, and 838, respectively. Baicalin inhibited bleomycin-induced apoptotic protein expression through the inhibition of TUNNEL-positive cells. Supporting this, baicalin at a concentration of 50 mg/kg promoted expression of Bcl-2 and suppressed Bax protein expression in rat lungs. Regarding the effect on lung fibroblasts, baicalin (80 µg/mL) suppressed the expression of cyclins A, D, and E, PCNA, p-AKT, and phosphorylated calmodulin-dependent protein kinase II (p-CaMKII). Furthermore, baicalin inhibited fibroblast proliferation by arresting cell-cycle progression at the G_0_/G_1_ phase, revealing that baicalin is a remarkable biomolecule for the treatment of PF.

The natural compound baccatin III (BAC), isolated from yew trees, has been reported to be effective in treating renal and hepatic inflammation and fibrosis [68]. D. Zhang et al. investigated its anti-fibrotic properties and underlying mechanisms in PF [69]. At a concentration of 5/10 mg/kg, BAC administration for 21 days significantly downregulated the Ashcroft and Masson’s scores in bleomycin-induced mice, in a dose-dependent manner. Through downregulation of the expression levels of the fibrosis markers α-SMA, fibronectin, collagens I and III, and TGF-β1, as well as reducing collagen deposition, the survival rate of mice was markedly enhanced. Moreover, a dose-dependent decline in inflammatory cell infiltration was observed, in addition to a reduction in the extent of pathological changes in the lungs of mice. Furthermore, it was confirmed that the attenuation of bleomycin-induced PF is mediated by inhibition of TGF-β1-expression, while suppression of TGF-β1 production occurred through the AKT/STAT6 signaling pathway. Administration of BAC inhibited TGF-β1-induced fibroblast differentiation via blocking the Smad signaling pathway. These data indicate that BAC attenuates PF effectively.

The anti-inflammatory and anti-proliferative molecule gambogic acid (GA) **22** (Table 1, entry 22) [70,71]—an active molecule of gamboge resin—was used to screen its effects against experimental PF [72]. The results indicated that E-cadherin was increased to 95% and vimentin was decreased to 113% with GA treatment at a 0.3 µM concentration, and the effect was reported to be dose-dependent. At concentrations of 0.15 µM and 0.5 µM, GA increased p-Smad3 expression to 157% and 148% respectively indicating that GA inhibits EMT via the TGF-β1/Smad3 pathway. Additionally, combination treatment with the Smad3 inhibitor SIS3 showed no significant differences compared to treatment with GA alone. In TGF-β1-stimulated HPMECs, GA treatment (0.5 µM) attenuated the decrease in VE-cadherin to 88% and blocked the increase in vimentin to 122%. These properties were found to be dose-dependent. GA administration modulated the rate of vasohibin-2 (VASH-2)/VASH-1 in HPMECs under hypoxic conditions. Alongside this, platelet-derived growth factor (PGDF) (111% at 0.5 µM) and fibroblast growth factor-2 (FGF-2) (114% at 0.5 µM) expression levels were significantly reduced in a dose-dependent manner, thereby inhibiting the proliferation of human lung fibroblast-1 (HLF-1) cells. The pathological score was reported to be decreased to 45% with GA treatment for 14 days, indicating inhibition of histopathological changes by GA in the early stages of PF. GA-treated rats demonstrated maximum reductions in hydroxyproline levels by 47% and in collagen contents by 55% at 1 mg/kg dose after 14 days. In addition, GA at a 1 mg/kg concentration decreased the expression levels of FGF-2, α-SMA, and PDGF, and the results were similar to those of the standard drug AP. Hence, gambogic acid deserves advanced investigation.

Owing to the preventive effects of sulforaphane (SFN) **23** (Table 1, entry 23) against bleomycin-induced PF via inhibition of oxidative stress [73], S. Y. Kyung et al. hypothesized that SFN might inhibit PF through inhibiting EMT [74]. At a concentration ≥40 µM, SFN reduced the viability of TGF-β1-induced MRC-5 cells by over 20%. The mRNA expression levels of fibronectin, collagens I and IV, and α-SMA in MRC-5 cells and fibroblasts were significantly decreased by the SFN treatment. SFN administration resulted in inhibition of Smad2/3 phosphorylation within 1 h, and this was maintained for 24 h. The bleomycin-induced histopathological changes were attenuated by SFN treatment, where the scores in SFN-treated mice were significantly lower than in the bleomycin group. Additionally, the hydroxyproline contents and the expression levels of the fibronectin and TGF-β1 proteins in the lungs were markedly ameliorated by SFN treatment. The E-cad-positive cells were found to be higher in number in the bleomycin + SFN group as compared with bleomycin group.

Tanshinone IIA **24** (Table 1, entry 24) (Tan IIA)—a bioactive molecule isolated from *Salvia miltiorrhiza* Bunge—has been reported to possess protective activity against lung injuries [75,76]. These findings led to investigation of the effects of Tan IIA against bleomycin-induced PF [77]. Through the intraperitoneal injection of Tan IIA, the pathological changes were significantly reversed, and aberrant collagen deposition in bleomycin-induced rat lungs was reduced by Tan IIA administration. Moreover, marked suppression of collagen I and hydroxyproline was observed with Tan IIA treatment. The immunofluorescence staining assay revealed that Tan IIA injection partially reversed the reduction in E-cadherin, while the expression levels of α-SMA, fibronectin, and vimentin were almost decreased to normal levels with Tan IIA treatment, which is indicative of the inhibitory effect of Tan IIA on enhanced EMT. Furthermore, Tan IIA inhibited the upregulation of pulmonary TGF-β1 messenger RNA and protein expression in bleomycin-instilled rats. The enhanced phosphorylation of Smad2 and Smad3 in bleomycin-treated rats was significantly reduced by Tan IIA, indicating that the anti-fibrotic effect of Tan IIA is because of inhibition of the TGF-β/Smad pathway. All of these inhibitory effects were also observed in in vitro experiments conducted using A549 cells.

### 1.3. Synthetic-Molecule-Based PF Inhibitors

Considering that HIF-1α plays an important role in the development of PF, while targeting HIF-1α can reduce PF [78], H. Huang et al. studied the in vitro and in vivo inhibitory effects of roxadustat **25** (Table 1, entry 25) on PF [79]. In this study, roxadustat-treated mice demonstrated improvements in inflammatory cell infiltration and thickening of the lung interstitium, followed by a significant decrease in pathology scores compared to bleomycin-treated mice. Compared to bleomycin-treatment mice (0.62%), a decrease in the lung coefficient (0.50%) in mice was attributed to the efficacy of the roxadustat treatment. This effect was found to be dose-dependent. A significant decline (from 26.3 mg/g to 19.0 mg/g) in HYP—a characteristic amino acid in collagen and an essential marker for collagen accumulation—indicated reduced collagen formation in roxadustat (40 mg/kg)-treated mice, followed by lower expression of collagens I and II. Additionally, roxadustat treatment reduced the protein expression of HIF-1α, prolyl hydroxylase domain-containing protein 2 (PHD2), α-SMA, p-Smad3, and CTGF in mice, thereby reducing the rate of oPF, but failed to inhibit TGF-β1 expression, as p-Smad3 and TGF-β1 are key factors for the PF pathway. Since complete inhibition of TGF-β1 expression was not observed, the drug roxadustat might have reduced PF through the inhibition of p-Smad3 expression.

Y. Zhou et al. investigated the effects of 1-(1-propanoylpiperidin-4-yl)-3-[4-(trifluoromethoxy)phenyl]urea (TPPU) **26** (Table 1, entry 26) against proliferation and differentiation of fibrosis in mice [80]. The decreased body weight in bleomycin-treated mice was restored by TPPU administration; furthermore, the mortality rate (55%) was reduced to 25% on day 14. The extent of pulmonary injury was decreased, including alveolar wall thickening and massive infiltration of inflammatory cells in the interstitium. TPPU (1 mg/kg/day) decreased collagen type I mRNA as compared with the bleomycin group, whereas there was no change in the expression of collagen type II mRNA. A significant reduction in the bleomycin-induced serum concentrations (*p* < 0.05) of IL-1β and IL-6 was noticed with TPPU instillation. Although TPPU could not inhibit the proliferation of murine fibroblasts at low concentrations (1 µM), at a concentration of 10 µM, a significant reduction in the proliferation of murine fibroblasts was observed. Additionally, at a similar concentration, TPPU decreased the percentage of S phase, indicating inhibition of fibroblast proliferation via the inhibition of DNA synthesis. Treatment of TGF-β1-sensitized murine fibroblasts with TPPU for 24 h resulted in decreased expression of the α-SMA mRNA and protein, revealing inhibition of fibroblasts’ differentiation into myofibroblasts. Furthermore, TPPU administration significantly increased the serum concentrations of 14- and 15-EET indicating that TPPU might constitute a potential approach for the treatment of PF.

The novel tyrosine kinase inhibitor ponatinib (PT) **27** (Table 1, entry 27) [81] was tested for its pharmacological activity against bleomycin-induced PF by Y. Qu et al. [82]. At a concentration of 0.3 µM, PT increased the E-cadherin expression level to 49.6% and decreased vimentin expression to 309% as compared with TGF-β1-stimulated A549 cells, indicating that PT attenuated alveolar EMT. PT (1 µM) decreased the apoptosis rate of alveolar epithelial type I (ATI) cells from 24.0% to 9.1%, and the anti-apoptotic effect was because of the decreased ratio of Bcl/Bax and RAGE expression in the stimulated ATI cells. PT (0.3 µM) inhibited HLF-1 proliferation in the presence of 50 µM suramin, while a significant effect was observed when treated with PT alone. Moreover, PT treatment for 14 days reduced the pathology scores by 45.6% in bleomycin-induced mice. Additionally, the collagen deposition was significantly decreased (from 5.60 to 3.01 mg/g tissue), as evident from the reduced hydroxyproline contents in lung tissues with PT treatment (1 mg/kg). Alongside these, PT treatment showed decreased expression levels of Smad3, p-Smad3, FGF-2, α-SMA, and PDGF-BB. These results suggest that ponatinib could be an excellent molecule for advanced investigation against PF.

As αvβ6 integrin plays crucial role in the activation of the probiotic factor TGF-β, A.E. John et al. explored GSK3008348 **28** (Table 1, entry 28) as an αvβ6 integrin inhibitor for the treatment of PF [83]. In this study, high binding affinity was observed between αvβ6 integrin and GSK3008348, thereby reducing TGF-β signaling to normal levels. Additionally, GSK3008348 demonstrated rapid internalization (t1/2 = 2.6 min) and lysosomal degradation of αvβ6 integrin. Furthermore, GSK3008348 decreased lung collagen deposition and serum C3M, reducing the progression of PF.

S. Song et al. explored the antiviral drug atazanavir sulfate (AS) for its protective effects against PF in vivo and in vitro [14]. In the CoCl_2_ group, hypoxic conditions showed decreased expression of E-cadherin and increased expression of vimentin, while AS administration increased E-cadherin expression by 117% and vimentin expression by 43.4%. In the test performed to check the effects of AS treatment, expression of HIF-1α, HMGB1, TLR-4, and p-NF-κB was found to be reduced when compared with the CoCl_2_ group, indicating that AS attenuates EMT via modulation of the HMGB1/TLR-4 pathway. Furthermore, AS was demonstrated to possess a protective effect mediated by HMGB1, where it decreased the HMGB1 expression while increasing the expression levels of aquaporin 5 and receptor for advanced glycation end products (RAGE) as compared with the CoCl_2_ group. AS treatment (1–10 µM) significantly inhibited HLF-1 cell proliferation in a concentration-dependent manner. Moreover, the combination treatment involving HCQ showed no further reduction in HLF-1 cell proliferation. The elevated pathology score was reduced by 51% with combined treatment with AS and ritonavir for 21 days as compared with the bleomycin group. Furthermore, the combination treatment markedly decreased the hydroxyproline content and collagen accumulation as compared with bleomycin-treated rats. These findings prove that the antiviral drug atazanavir could be repurposed as anti-fibrotic drug.

Apart from small-molecule inhibitors, the 29-amino-acid synthetic peptide PD-29—with the properties of anti-angiogenesis, matrix-metalloproteinase-inhibitory activity, and inhibition of integrin—has been investigated for its effects against PF in bleomycin-induced rats [84]. PD-29 (7.5 mg/kg and 5 mg/kg) administration greatly enhanced the survival rates (100%) and was found to be more effective than nintedanib (91.6%). Administration of PD-29 and nintedanib resulted in significant reductions in the serum levels of TGF-β1 and IL-6. Moreover, PD-29 exhibited strong antioxidant properties, as evident from the modulation of malondialdehyde (MDA) and GSH levels. The integrins (αvβ3 and αE) responsible for the activation and release of TGF-β were decreased remarkably with PD-29 treatment when compared with the model group. The imbalance of matrix metalloproteinase/tissue inhibitor of metalloproteinase (MMP/TIMP) that causes lung tissue damage was regulated by PD-29 administration. The factors associated with angiogenesis (e.g., VEGF-A, FGF-2, and PDGF-B) were reduced in a dose-dependent manner, and significant differences were observed when compared with the model group. Alongside this, the elevated levels of phosphorylated Smad2 and Smad3 were inhibited by PD-29 and nintedanib treatment, while upregulation of Smad7 was observed in bleomycin-instilled rats. At a concentration of 10 μmol/L, PD-29 inhibited expression of α-SMA and Col I while promoting the expression of E-cad. Furthermore, PD-29 was reported to be non-toxic in A549 cells as well as in murine primary lung fibroblasts at concentrations of 0.1–100 μmol/L. This innovative research reveals that synthetic peptides could be promising anti-fibrotic agents to treat PF.

A low-molecular-weight thiol, pyrrolidine dithiocarbamate (PDTC) **29** (Table 1, entry 29), has been demonstrated to attenuate the development of monocrotaline-induced pulmonary arterial hypertension via inhibition of NF-κB [85], which relates to the pathogenesis of lung diseases. With this in mind, M.A. Zaafan et al. evaluated the efficacy of PDTC against bleomycin-induced PF [86]. In this study, pretreatment of rats with PDTC (38.90 ng/g wet tissue) decreased the elevated hydroxyproline contents in bleomycin-treated rats, and the results were comparable to those of normal control rats. Then, the profibrotic factors TGF-β1 and TNF-α were significantly reduced to 34.83 pg/g and 7.23 pg/g, respectively. Compared to bleomycin-induced rats, PDTC led to reductions in the elevated MDA (99.69 µM/g) and nitrite (73.47 µM/g) contents, and a subsequent increase in GSH content was observed (6.09 mg/g wet tissue). PDTC treatment (100 mg/kg) restored the total leukocyte counts to almost normal levels (0.68 × 10^6^ mL^−1^). Additionally, the percentages of neutrophils and eosinophils were reduced, whereas the percentages of lymphocytes and alveolar macrophages were increased as compared with the bleomycin group. Alongside this, PDTC administration demonstrated moderate immunohistochemical expression of iNOS as compared to the bleomycin group. At a concentration of 100 mg/kg, inflammatory cell infiltration was significantly reduced (0.92) compared to the bleomycin-induced group (3.50). Masson’s trichrome staining indicated a marked decrease (0.95) in the collagen deposition in PDTC-treated rats. Furthermore, focal fibrosis was reported to be reduced to 0.60. Overall, PDTC could be developed into a good anti-fibrotic agent.

Amitriptyline (ATP) **30** (Table 1, entry 30), possessing anti-inflammatory [87], antioxidant [88], and antidepressant properties, was investigated for its effect on the inhibition of PF [89]. In this study, ATP treatment showed marked suppression of elevated hydroxyproline and TGF-β1 contents. Alongside this, restoration of normal histological structure, with remarkable decreases in inflammatory cell infiltration and fibrosis, and the suppression of collagen deposition and fibroblastic cell proliferation, was observed when compared to the bleomycin-treated group. ATP administration resulted in significant decreases in pro-inflammatory and inflammatory mediators such as NF-кβ, TNF-α, and iNOS. Moreover, through the elevation of the pulmonary contents of Nrf2 and GSH, and the marked suppression of the lipid peroxide contents in lung tissue, ATP demonstrated suppression of oxidative stress as compared with the bleomycin-instilled group. Furthermore, ATP treatment reduced the amount of α-SMA and the expression of p53 in lung tissue. However, no significant change was reported in terms of the pulmonary contents of serpine-1 when compared between the bleomycin- and ATP-treated groups.

Soluble epoxide hydrolase (sEH) inhibitors have been demonstrated to modulate the levels of epoxy fatty acids in tissues, thereby reducing acute lung inflammation [90] and attenuating fibrosis and inflammation in the liver, heart, kidneys, and pancreas [80]. In this regard, X.-W. Dong et al. investigated the inhibitory activity of the sEH inhibitor 12-(3-adamantan-1-yl-ureido)dodecanoic acid (AUDA) **31** (Table 1, entry 31) against bleomycin-induced pulmonary toxicity [3]. Initially, AUDA administration (10 mg/kg) resulted in a decline in the body weight loss in bleomycin-treated mice after 3 and 4 weeks. Inflammatory cell infiltration of macrophages, neutrophils, and lymphocytes in the lungs was attenuated by AUDA treatment. Moreover, expression of IL-1β, TGF-β1 m, and MMP-9 mRNA was significantly reduced by AUDA at a concentration of 10 mg/kg. Masson staining indicated that AUDA alleviated the collagen deposition in the alveolar space and the tela submucosa. Furthermore, this was proven by the reduction in the hydroxyproline contents in the lung tissues as compared with mice treated with bleomycin alone. Meanwhile, AUDA administration (3 or 10 mg/kg) decreased the expression of E-cadherin and α-SMA mRNA, suggesting that the mechanism involved is the suppression of phosphorylation of the p38 and Smd3 proteins. Since AUDA is an sEH inhibitor, it was explored for the inhibition of TGF-β1-induced sEH expression, where the sEH protein expression in 16HBE cells was found to be reduced at a concentration of 10 µM. Furthermore, AUDA at a concentration of 10/20 µM regulated EMT via inhibition of p38 and Smad2/3 activation.

The anti-HIV drug raltegravir was shown to decrease the degree of hepatic steatosis in HIV-infected patients with nonalcoholic fatty liver disease [91]. Considering this fact, X. Zhang et al. studied the anti-fibrotic effects of raltegravir in vitro and in vivo [92]. The increased lung coefficient of bleomycin-induced mice was reduced by raltegravir (40, 80, 120 mg/kg/day), and this effect was found to be dose-dependent. On day 35, expression of collagens I and II and hydroxyproline contents were reported to be ameliorated by raltegravir treatment. The protein expression of nucleotide-binding leucine-rich pyrin 3 (NLRP3), HMGB1, TLR4, p-NF-κB, HIF-1α, and α-SMA was decreased in the raltegravir-treated group. In addition, the fibroblast marker expression was found to be attenuated (*p* < 0.01) compared to bleomycin-treated mice. Furthermore, at a concentration of 10 µM, raltegravir inhibited L929 cell proliferation and attenuated experimental lung fibrosis by blocking NLRP3/HMGB1/TLR4 signaling. These results indicate that raltegravir is a potent anti-fibrotic agent.

Azithromycin (AZT) has been reported to possess anti-fibrotic effects, in addition to antibiotic properties [93,94]. In this regard, K. Krempaska et al. studied the anti-fibrotic effect of AZT on fibroblasts in comparison with a control group, and then they investigated the underlying mechanisms of cell death, autophagy, and lysosomal function [95]. After treatment of IPF fibroblasts with AZT at a concentration of 50 µM for 24 h, gene expression of the profibrotic marker collagen Iα1 was significantly decreased, while in the analysis of individual IPF fibroblasts and controls, fibronectin was reduced significantly. Moreover, the AZT treatment resulted in decreased protein expression of αSMA in IPF fibroblasts. In the analysis of protein expression of autophagic markers such as microtubule-associated protein light chain 3 (LC3I), LC3II, and p62, AZT treatment demonstrated an increased LC3II/I ratio, indicating decreased autophagic flux and, in turn, decreased autophagic activity. Furthermore, increased p62 protein expression in IPF fibroblasts indicated reduced autophagy compared to controls. Stimulation with AZT and TGF-β enhanced the early apoptosis in IOF fibroblasts compared to TGF-β alone, and this effect was also proven by the reduced anti-apoptotic activity of Bcl-xL. AZT treatment impaired the gene expression of lysosomes, as evident from the downregulation of the Ras-related protein Rab-7b (RAB7B). Furthermore, AZT was reported to interfere with lysosomal function through lysosomal accumulation in IPF fibroblasts as well as controls. These facts indicate that AZT might be an efficient therapeutic agent against PF after confirming its anti-fibrotic effects through in vivo tests.

### 1.4. Miscellaneous

Recent studies have reported that arsenic trioxide has been used as a drug for ulcers, malaria, and psoriasis [96], and it has been reported to regulate signaling pathways in cancer cells [97]. These facts inspired Luo et al. to investigate the inhibitory activity of arsenic trioxide against TGF-β1-induced fibroblast-to-myofibroblast differentiation [98]. Initially, at a concentration of 10 nM, arsenic trioxide demonstrated a twofold reduction in the expression of TGF-β1-induced α-SMA (14.76; *p* < 0.05) and type I collagen (9.10; *p* < 0.05). Moreover, pretreatment with arsenic trioxide reduced TGF-β1-induced expression of CTGF and plasminogen activator inhibitor-1 (PAI-1) mRNA. TGF-β1 reduced collagen I gel (78%; *p* < 0.05), thereby increasing contractility, while treatment with arsenic trioxide at 10 nM enhanced the collagen I gel (89%; *p* < 0.05). Then, the inhibitory effect of arsenic trioxide on Smad2/3 was investigated at 10 nM, where arsenic trioxide decreased the level of phosphorylation of Smad2/3, and this effect was found not to be due to the chemical inactivation of the TGF-β1. Alongside this, arsenic trioxide pretreatment attenuated TGF-β1-mediated Akt phosphorylation. However, further studies revealed that arsenic trioxide does not globally affect phosphorylation. At a 10 nM concentration, arsenic trioxide showed a blocking effect on TGF-β1-induced H_2_O_2_ expression (1.11; *p* < 005) and upregulation of NADPH oxidase 4 (NOX-4) mRNA (119.50; *p* < 0.05). More importantly, downregulation of PML proteins was observed with arsenic trioxide pretreatment for 24 h at a 20 nM concentration. Particularly, in a study of lung fibrosis inhibition in C57Bl/6 mice, arsenic trioxide (1 mg/kg) reduced PML body intensity in vivo (0.32 and 0.37; *p* < 0.05). A noteworthy decline in the collagen I mRNA expression (3.61; *p* < 0.05) was observed, and downregulation of α-SMA mRNA (0.84; *p* < 0.05) was evident after arsenic trioxide pretreatment. All of these findings indicate that arsenic trioxide should be investigated further to explore its efficacy.

H. Jiao et al. explored the effects of sodium arsenite (SA) against lung fibroblast differentiation and PF [99]. In this study, SA had no effect on FBS-induced cell proliferation and cell death. However, at a concentration of 10 µM/L, SA reduced the TGF-β-instilled expression of α-SMA in cultured fibroblasts. Moreover, intracellular reactive oxygen species (ROS) in normal human cell fibroblasts (NHLFs) induced by TGF-β were reported to be inhibited. Smad3 phosphorylation and ERK phosphorylation were significantly reduced by SA pretreatment. Administration of SA (2 mg/kg) in bleomycin-induced mice after 21 days significantly reduced the lung injury, and then the collagen protein expression and subsequent collagen deposition in lung tissues were suppressed by the SA treatment. In addition, the expression of fibrotic markers such as FN and Col1a2 was downregulated in lung tissue by the SA administration.

Innovatively, L. Gao et al. explored the inhalation of H_2_ in reducing the pathogenesis of PF [93,94], considering the antioxidant properties of H_2_ that protect tissues such as the liver, kidneys, and lungs from damage [100]. H_2_ inhalation for 2, 4, or 8 h per day suppressed the expression of TGF-β1 and TNF-α at the mRNA and protein levels, indicating amelioration of profibrogenic cytokines. For a duration of 8 h, H_2_ inhalation restored the reduced E-cadherin and elevated α-SMA expression levels to normal levels. The factors associated with pathogenesis in bleomycin-induced mice—such as thickening of the alveolar septum, disappearance of the alveolar cavity, distortion of lung structure, and extensive inflammatory cell infiltration—were significantly reduced by H_2_ inhalation. Then, extensive deposition of fibrillar collagen was alleviated by H_2_ inhalation. The oxidative stress was found to be mitigated through reduced formation of ROS, MDA, and hydroxyproline in bleomycin-treated mice. Furthermore, the genes linked to fibroblast differentiation and collagen accumulation—such as α-SMA, collagen type-1 alpha-1 (COL1A1), and TGF-β—were markedly downregulated by H_2_ inhalation. Above all, inhalation of H_2_ may potentially attenuate PF.

### 1.5. Chronic Obstructive Pulmonary Disease (COPD)

COPD is an obstructive lung disease associated with long-term breathing problems and poor airflow [101] that worsen over time [100]. COPD is also known as chronic bronchitis or emphysema. The most general symptoms include shortness of breath and cough with sputum production [101]. The main cause of COPD is tobacco smoking, and a few cases may be because of air pollution and/or genetics [102]. People over the age of 40 years may become victims of this disease. Approximately 2.4% of the global population was affected by COPD as of 2015, at which time 90% of the deaths were recorded in developing countries [103]. Preventive measures for most COPD cases include cessation of smoking and improving indoor and outdoor air quality [95]. When it comes to treatment, the progress of the disease can only be slowed down—a complete cure is still unavailable [104,105]. Amongst the known medications, corticosteroids are usually given to decrease acute exacerbations in patients with either moderate or severe disease [106]. However, inhalation of steroids is associated with increased rates of pneumonia [107] and, furthermore, long-term treatment with steroid tablets would result in significant side effects [108]. Clinical trials are underway to increase the efficiency of the drugs used in the treatment of COPD. Therefore, there is an urgent need to discover and develop potent drugs to prevent COPD. Herein, the drug molecules identified in recent years for the amelioration of COPD are discussed.

### 1.6. Inhibition of Cigarette Smoke (CS)-Induced COPD

Ginsenoside Rg1—a major constituent of medicinal herb *Panax ginseng*—has been demonstrated to inhibit EMT in hepatic carcinoma and renal tubulointerstitial fibrosis via TGF-β1 suppression [107]. S. Guan et al. studied the mechanism involved in the inhibition of CS-induced pulmonary remodeling [108]. CS induction increased (19.64%) pulmonary interstitial fibrosis, which was attenuated (8.81%) by ginsenoside Rg1 treatment at a concentration of 20 mg/kg, indicating inhibition of CS-induced emphysema. Ginsenoside Rg1 treatment restored the decreased α-SMA and vimentin expression levels. Additionally, ginsenoside Rg1 reduced the TGF-β1 levels (47.55) in serum and downregulated the mRNA and protein expression of TGF-β1 as compared with the COPD group. Moreover, the cell viability of human bronchial epithelial (HBE) cells was decreased by 13.2% when treated with ginsenoside Rg1 at a concentration of 80 µM. Additionally, elevated expression levels of α-SMA (1.07) and reduced E-cad levels (0.55) were restored to their normal levels of 0.70 and 0.74, respectively, by ginsenoside Rg1 administration. Smad2/3 activation was inhibited by the drug treatment, thereby diminishing the signaling pathway. All of the inhibitory properties of ginsenoside Rg1 are dose-dependent. The results indicate that ginsenoside Rg1 can alleviate CS-induced COPD via the TGF-β1/Smad pathway.

So-Cheong-Ryoung-Tang (SCRT)—a mixture of eight herbal preparations—is a traditional oriental medicine [109]. Reports indicate that it can attenuate the production of cytokines and chemokines for airway remodeling in allergic asthma. In this regard, N. Shing et al. explored the anti-inflammatory effects of CSRT on airway inflammation induced by CS and lipopolysaccharides [110]. The elevated expression levels of pro-inflammatory cytokines such as TNA-α, IL-6, and IL-1β were reduced by SCRT administration. Moreover, SCRT demonstrated a marked reduction in iNOS, along with phosphorylation of NF-κB expression. Increased numbers of inflammatory cells in bronchoalveolar lavage fluid (BALF) were reported following CS and LPS induction, which were abated in a dose-dependent manner by the SCRT treatment. Alongside this, inflammatory cell infiltration was significantly alleviated in a dose-dependent manner in SRCT-treated mice as compared with CS- and LPS-exposed mice. Furthermore, the protease responsible for physiological and pathological processes was reduced by SCRT administration.

The natural polyphenolic flavonoid silymarin **32** (Table 1, entry 32) (with the active component silibinin), obtained from *Silybum marianum*, has been reported to possess anti-inflammatory and anti-fibrotic activity [111]. J. Ko et al. explored silibinin for its anti-fibrotic effects against CS-induced COPD [112]. Silibinin-treated (10 and 20 mg/kg) mice exhibited reduced inflammatory cell numbers in BALF compared to the CS group. The levels of CS-induced cytokines such as IL-6 and TNF-α were markedly attenuated by silibinin treatment. Then, silibinin reduced the inflammatory cell infiltration in lung tissues. The effect was similar to that of roflumilast treatment. Alongside this, extensive collagen deposition was significantly attenuated in a dose-dependent manner in the silibinin group compared to the CS + LPS group. The expression levels of TGF-β1 and collagen were remarkably decreased with silibinin treatment. Silibinin-treated mice also exhibited decreased phosphorylation of Smad 2/3.

Growing evidence indicates that traditional polyherbal mixture Mahuang-Tang (MT) possesses anti-inflammatory properties, in addition to other potent pharmacological activities [112]. With this in mind, J. Ko et al. investigated the suppressive effects of Mahuang-Tang water extract (MTWE) against CS-induced anti-inflammatory responses in COPD [113]. In the initial cytotoxic study, MTWE exhibited no toxic effect on H292 cells at concentrations up to 20 µg/mL. Pro-inflammatory cytokines such as IL-1β, IL-6, and TNF-α, along with their mRNA expression levels, were decreased by the MTWE treatment compared to controls. In support of this, MTWE treatment at a concentration of 100 mg/kg showed a twofold reduction in inflammatory cell numbers (246.7). This effect was found to be dose-dependent. Furthermore, Erk phosphorylation and MMP-9 expression levels were markedly reduced by MTWE administration. However, the efficacy may increase only if the individual components of the herbal medicine are separated and subsequently investigated for anti-fibrotic effects.

Because of the potent anti-inflammatory responses of *Callicarpa japonica* Thunb. (CJT), it has been investigated for its effects against CS-induced COPD [114]. CJT showed a marked decrease in the numbers of neutrophils in BALF compared with the COPD group, in a dose-dependent manner. CS-induced production of mediators of reactive oxygen species (ROS) and BALF was significantly attenuated by CJT treatment. At a concentration of 30 mg/kg, CJT alleviated pro-inflammatory cytokines such as TNF-α and IL-6 in BALF. Compared with COPD mice, CJT-treated mice demonstrated decreased production of inflammatory cells and mucus. Moreover, CJT showed a dose-dependent inhibition of the mRNA expression of mucin 5AC (MUC5AC) and cytokines in H292 cells. Furthermore, pretreatment of mice with CJT attenuated the phosphorylation of ERK in H292 cells. These facts indicate that CJT has potent activity against COPD.

N. Shin et al. studied the attenuation effects of Galgeun-tang water extract (GTWE) against CS- and LPS-induced pulmonary inflammation [115]. In this study, GTWE (50 mg/kg) reduced inflammatory cell infiltration into the lung tissue as compared with the CS group. Meanwhile, at a higher dose, GTWE exhibited similar results compared to the roflumilast-treated group. In addition, a significant reduction in CS- and LPS-induced inflammatory mediators in BALF was observed following GTWE treatment. The CS group showed increased expression of iNOS and cyclooxygenase-2 (COX-2), and these effects were reversed in the GTWE-treated groups. Then, the marked reduction of phosphorylation of IκBα and NF-dB. in lung tissue was evident after GTWE instillation.

N. Shin et al. investigated the effects of melatonin **33** (Table 1, entry 33) on CS- and LPS-induced PF [116], considering the potent antioxidant and anti-inflammatory activity of melatonin [117]. Melatonin administration (20 mg/kg) significantly reduced the neutrophil count to 405.1 as compared with CS- and LPS-induced mice (583.4). Alongside this, marked decreases in the IL-6 and TNF-α levels in BALF were observed. Reduced collagen deposition was also observed in melatonin-treated mice compared to CS- and LPS-exposed mice. Moreover, melatonin decreased TGF-β1 expression and inhibited Smad3 phosphorylation. Additionally, melatonin exhibited reduced collagen I expression. Furthermore, melatonin instillation suppressed the expression of profibrotic mediators such as IL-6 and TNF-α in CSC-stimulated H292 cells. Finally, melatonin could also decrease the expression of collagen I and phosphorylation of Smad2/3 and p38. These results indicate that melatonin could be a good anti-fibrotic agent.

## 2. Discussion

PF is a condition (rather than a disease) wherein the soft lung tissues is converted into a fibrous material, eventually leading to reduced oxygen uptake capacity. During this stage, various physiological changes take place in the lungs, as well as elsewhere in the body. In order to understand the actual mechanisms of PF, researchers use bleomycin to induce PF in mice. The changes that take place in the bleomycin-induced mice are noted, where increases in the mass of the lungs and reductions in the body mass are the most prevalent effects. The increase in the mass of the lungs is due to the accumulation of extracellular matrix and collagen in large amounts. Sometimes, the hydroxyproline content in the lungs is considered as a marker of collagen accumulation. The experimental results prove that epithelial-to-mesenchymal transition (ETM) is associated with tissue fibrosis, producing numerous ECM proteins [118]. The pro-inflammatory factor TNF-α is also responsible for the promotion of ECM [119]. Moreover, significant histopathological changes are observed in the bleomycin-induced mice, including infiltration of inflammatory cells, destruction of the lung architecture, and disruption of alveolar units. Furthermore, myofibroblast differentiation is a serious issue in PF [120], wherein α-SMA is an active indicator that indicates the presence of active fibrogenic cells. These are the consequences of the probiotic factor TGF-β1, which is often found to be elevated [121,122,123]. Hence, the inhibition of TGF-β1 is a good approach for the alleviation of PF. Other consequences of increased TGF-β1 include production of ROS and suppression of antioxidant enzymes, eventually resulting in redox imbalance [124]. This shows that TGF-β1 can regulate oxidative stress in the process of tissue repair, leading to fibrosis. Most of the anti-fibrotic agents discussed to date inhibit the elevated TGF-β1 levels induced by bleomycin. The literature shows that elevated phosphorylation of Smad2/2 and extensive production of cytokines responsible for fibrosis are observed in PF. In addition, hypoxia is one of the important issues that arise due to structural degeneration of lung tissue, which needs to be addressed; it may even lead to death of the patient if untreated [125]. Inhibition of the TGF-β1/Smad signaling pathway is hypothesized as the most acceptable pathway for the inhibition of PF. When it comes to treatment, except for some promising anti-fibrotic agents such as nintedanib and pirfenidone that can reduce the progression of PF, no efficient synthetic drug has been discovered for PF to date. However, there are a few drugs that have been repurposed to prevent the progression of PF. In view of this, it is necessary to shed some light on the progression of drug discovery and development for the attenuation of PF. In this regard, we have attempted to comprehensively discuss the drugs reported for the alleviation of PF so far. The research in this field reveals that many natural/biomolecule-based drugs are active against PF and possess enough potential to be developed into promising inhibitors of PF. Similarly, COPD may be induced by many factors, including CSe, which is a primary risk factor for the development of COPD. COPD also one of the potent health issues for which drug discovery should be taken up on a priority basis. In this review, short descriptions of recently reported drugs for COPD are provided.

## 3. Future Perspectives

In this study, most of the pharmacological evaluation against PF and COPD was performed on naturally available molecules/biomolecules. There is no evidence of any comparative study of a set of molecules for the inhibition of PF or COPD. However, growing evidence suggests that amongst a set of molecules tested for biological activity, one, two, or a few molecules may be potent molecules. The same phenomenon might apply even in the case of inhibition of PF. Researchers should investigate the inhibitory effects of the structural derivatives of the reported molecules. Alongside this, molecular docking simulation studies of a series of molecules would aid in arriving at probable conclusions with respect to pharmacological activity, which is essential for further drug development. In addition, cytotoxicity studies are the most important activity to consider in terms of drug efficacy. Furthermore, to increase the bioavailability and improve the metabolic stability of the drug, encapsulation may be required, which would reduce the enzymatic degradation of the drug in vivo.

## Figures and Tables

**Figure 1 molecules-28-03674-f001:**
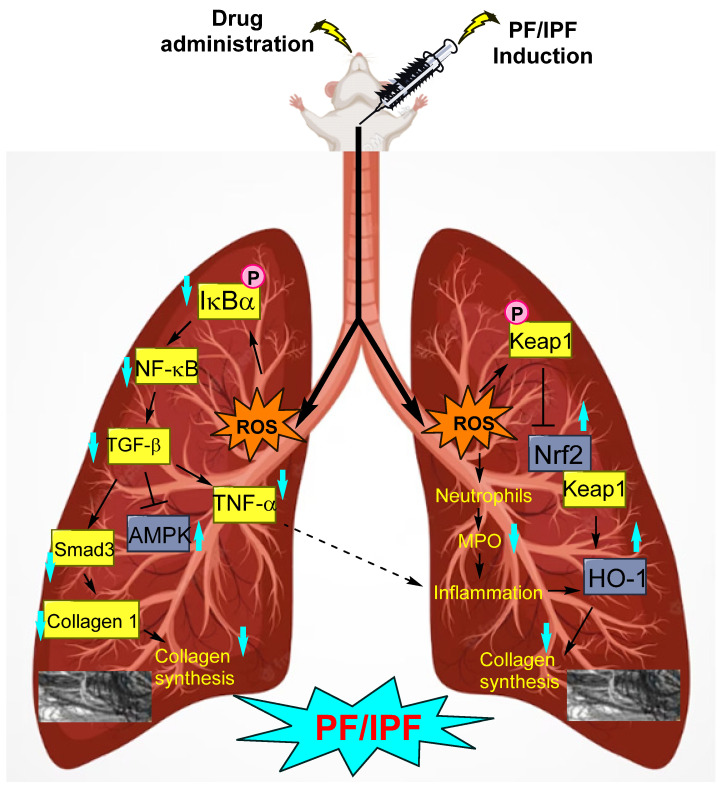
Illustration of the pathways of biomarkers that lead to PF/IPF (adapted from Zhou et al. [4]).

**Table 1 molecules-28-03674-t001:** Biomolecule-based inhibitors’ structures, common names, IUPAC names, and CAS numbers.

Entry	Structure	Number in the Text	Common Name	IUPAC Name	CAS Number
1	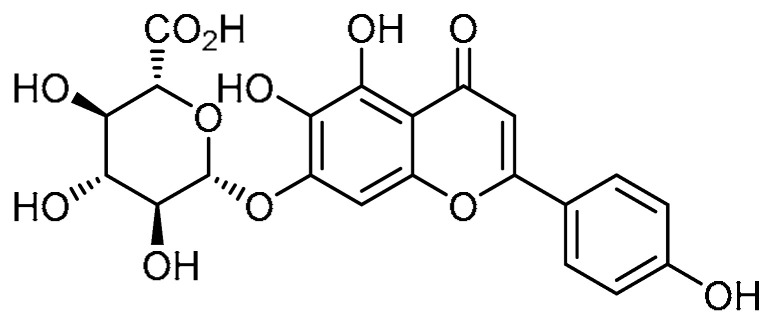	**1**	Scutellarin	4′,5,6-Trihydroxyflavon-7-yl β-D-glucopyranosiduronic acid	27740-01-8
2	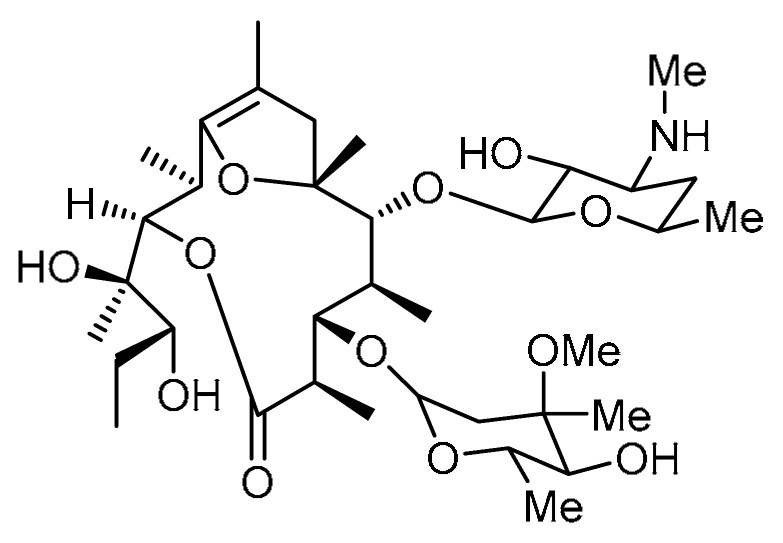	**2**	Erythromycin 703	(2R,3R,6R,7S,8S,9R,10R)-3-((2R,3R)-2,3-Dihydroxypentan-2-yl)-7-(((4R,5S,6S)-5-hydroxy-4-methoxy-4,6-dimethyltetrahydro-2H-pyran-2-yl)oxy)-9-(((2S,4S,6R)-3-hydroxy-6-methyl-4-(methylamino)tetrahydro-2H-pyran-2-yl)oxy)-2,6,8,10,12-pentamethyl-4,13-dioxabicyclo [8.2.1]tridec-1(12)-en-5-one	
3	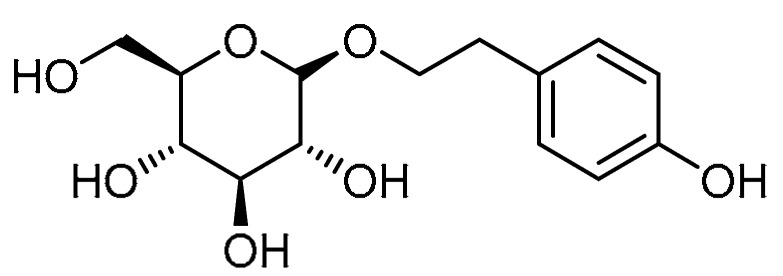	**3**	Salidroside	2-(4-Hydroxyphenyl)ethyl β-D-glucopyranoside	10338-51-9
4	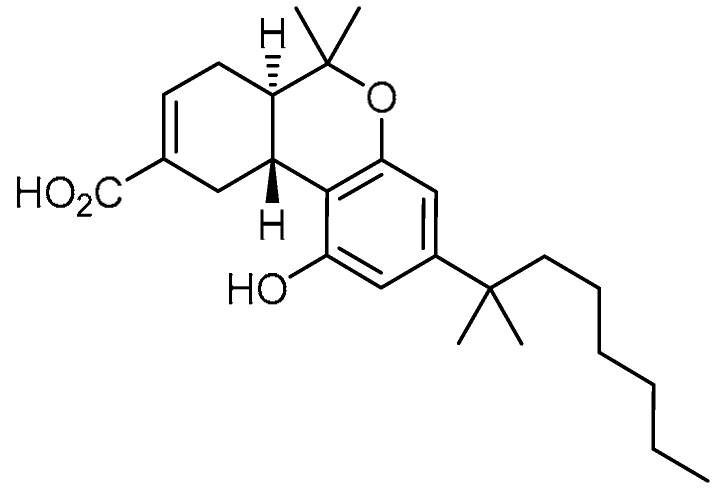	**4**	Ajulemic acid	(6aR,10aR)-3-(1,1-Dimethylheptyl)-6a,7,10,10a-tetrahydro-1-hydroxy-6,6-dimethyl-6H-dibenzo[b,d]pyran-9-carboxylic acid	137945-48-3
5	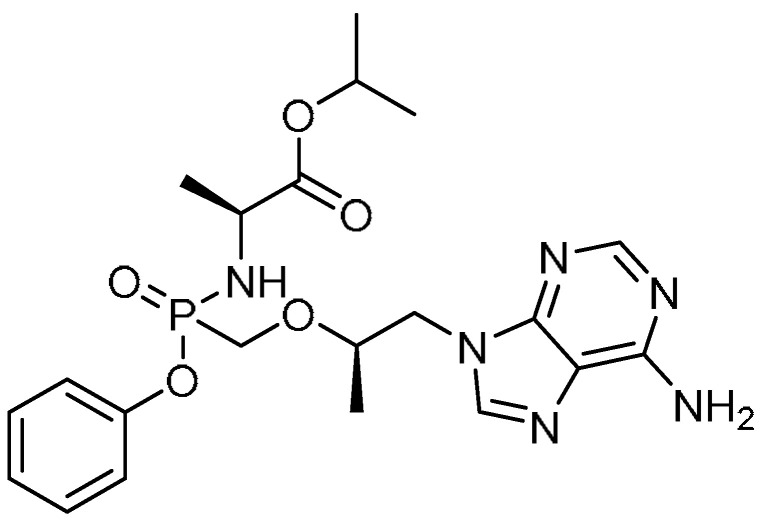	**5**	Tenofovir alafenamide fumarate	Isopropyl (2S)-2-[[[(1R)-2-(6-aminopurin-9-yl)-1-methyl-ethoxy]methyl-phenoxy-phosphoryl]amino]propanoate	379270-37-8
6	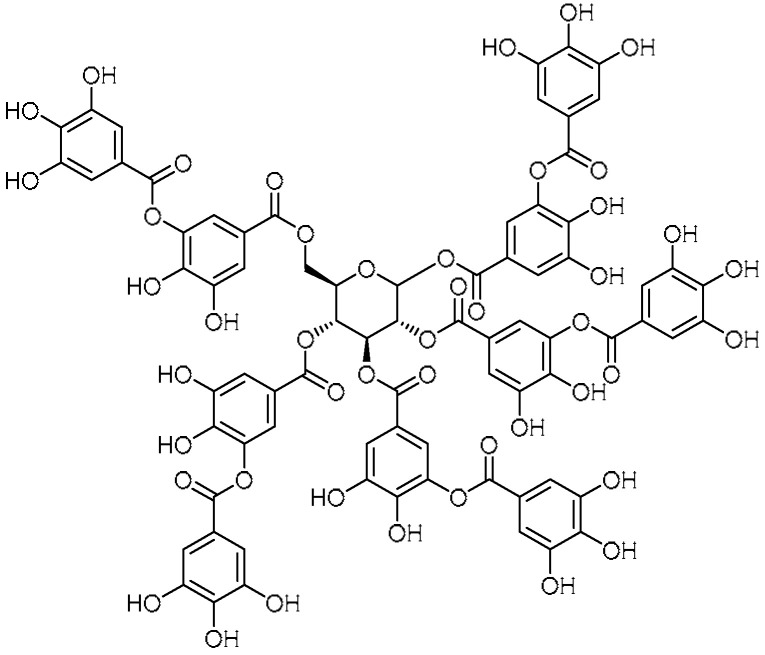	**6**	Tannic acid	1,2,3,4,6-Penta-O-{3,4-dihydroxy-5-[(3,4,5-trihydroxybenzoyl)oxy]benzoyl}-D-glucopyranose	1401-55-4
7	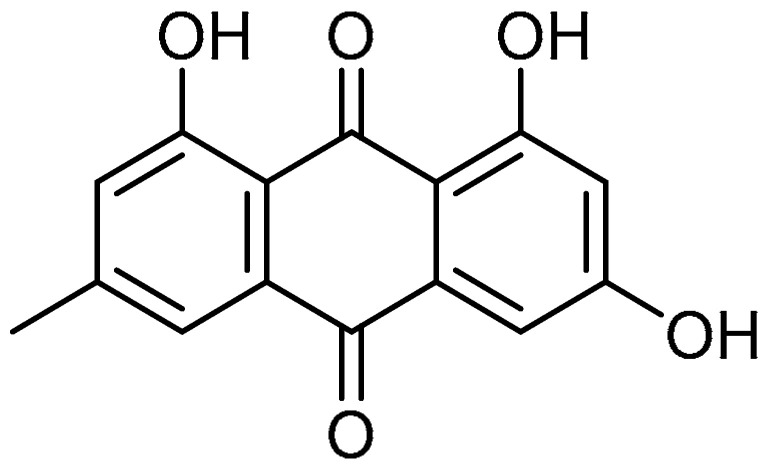	**7**	Emodin	1,3,8-Trihydroxy-6-methylanthracene-9,10-dione	518-82-1
8	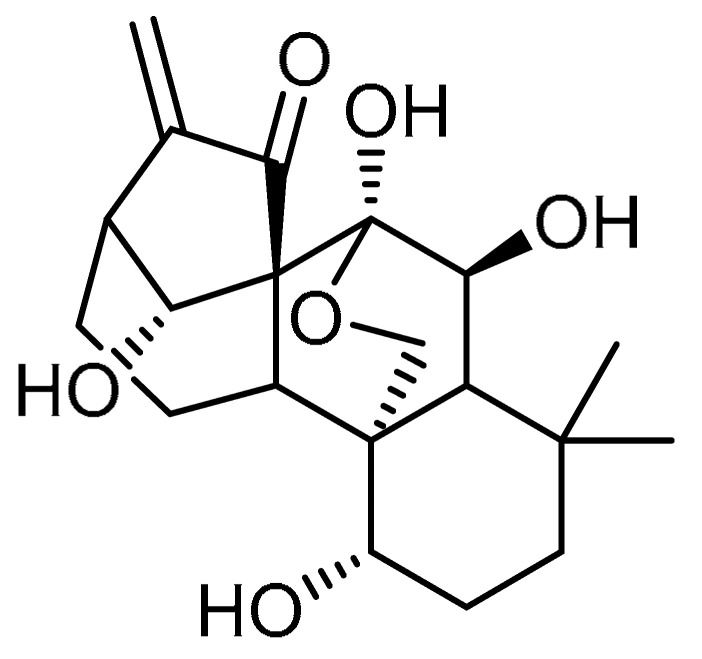	**8**	Oridonin	7a,20-Epoxy-1a,6b,7,14-tetrahydroxy-Kaur-16-en-15-one	28957-04-2
9	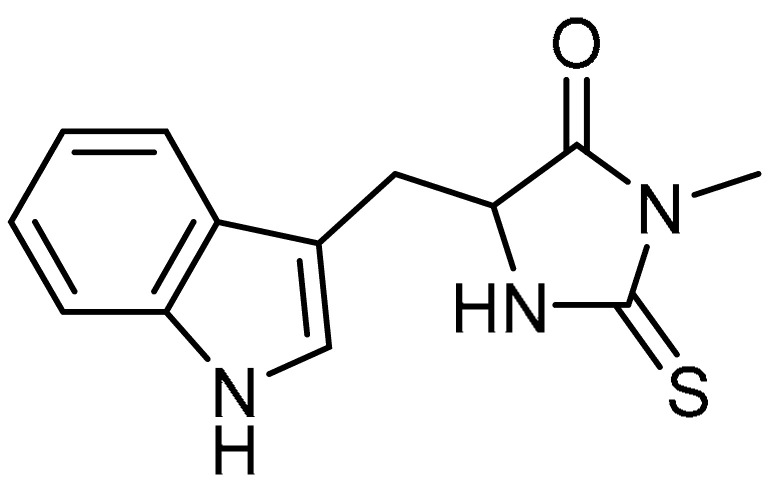	**9**	Necrostatin-1	5-(1H-Indol-3-ylmethyl)-3-methyl-2-thioxo-4-Imidazolidinone	4311-88-0
10	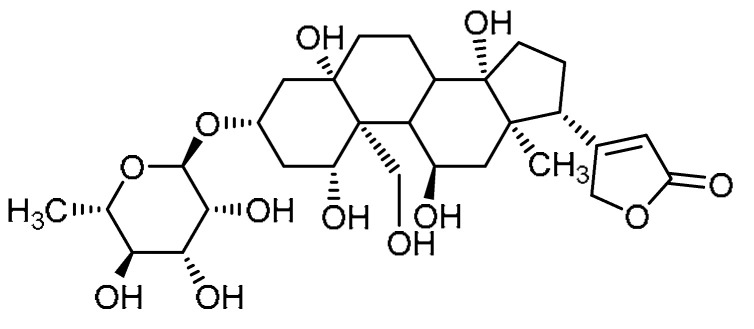	**10**	Ouabain	1β,3β,5β,11α,14,19-Hexahydroxycard-20(22)-enolide 3-(6-deoxy-α-L-mannopyranoside)	630-60-4
11	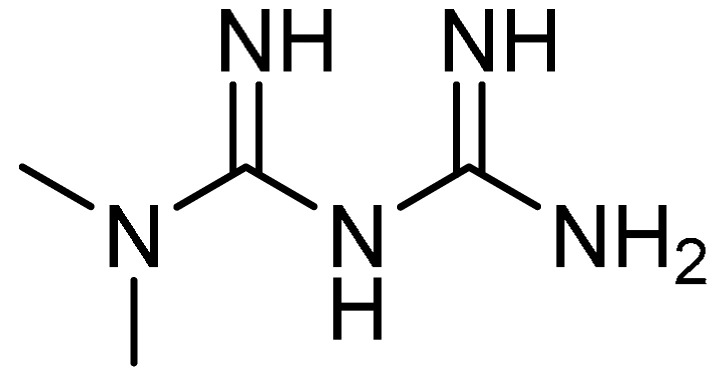	**11**	Metformin	N,N-Dimethylimidodicarbonimidic diamide	657-24-9
12	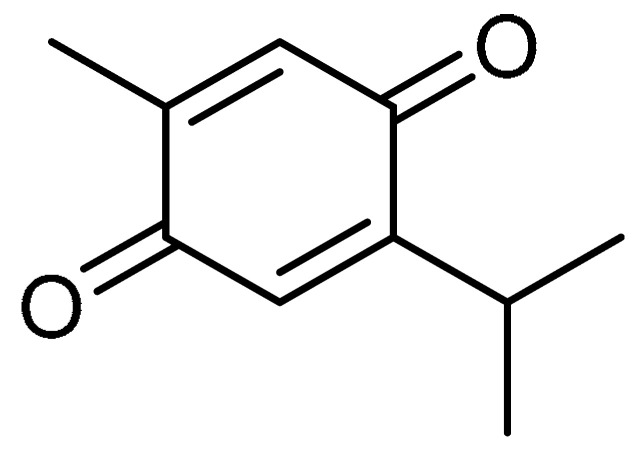	**12**	Thymoquinone	2-Methyl-5-(propan-2-yl)cyclohexa-2,5-diene-1,4-dione	490-91-5
13	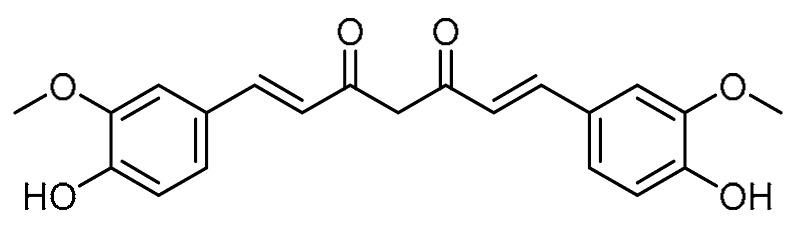	**13**	Curcumin	(1E,6E)-1,7-Bis(4-hydroxy-3-methoxyphenyl)hepta-1,6-diene-3,5-dione	458-37-7
14	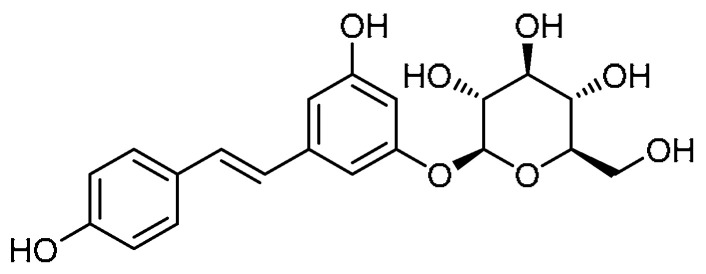	**14**	Polydatin	(2S,3R,4S,5S,6R)-2-{3-Hydroxy-5-[(E)-2-(4-hydroxyphenyl)ethen-1-yl]phenoxy}-6-(hydroxymethyl)oxane-3,4,5-triol	27208-80-6
15	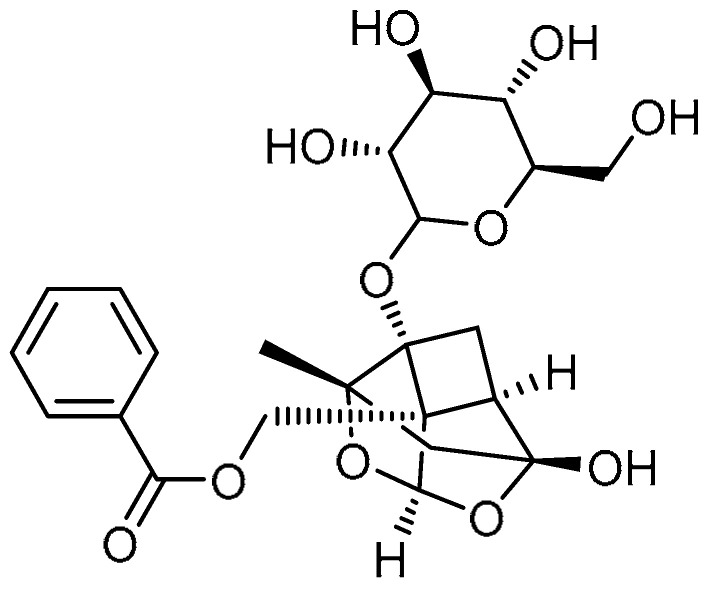	**15**	Paeoniflorin	((1aR,1a1S,2S,3aR,5R,5aR)-5-Hydroxy-2-methyl-1a-(((3R,4S,5S,6R)-3,4,5-trihydroxy-6-(hydroxymethyl)tetrahydro-2H-pyran-2-yl)oxy)tetrahydro-1H-3,4-dioxa-2,5-methanocyclobuta[cd]pentalen-1a1(3aH)-yl)methyl benzoate	23180-57-6
16	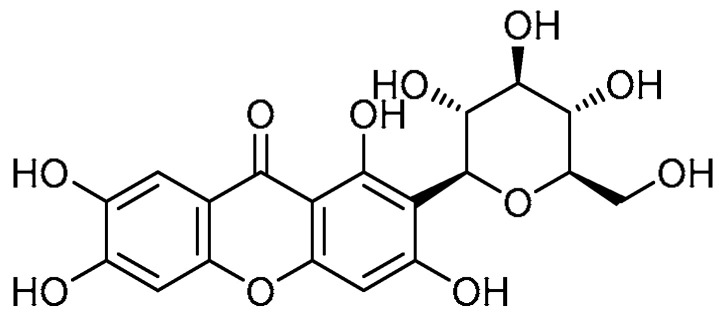	**16**	Mangiferin	1,3,6,7-Tetrahydroxy-2-[(2S,3R,4R,5S,6R)-3,4,5-trihydroxy-6-(hydroxymethyl)oxan-2-yl]-9H-xanthen-9-one	4773-96-0
17	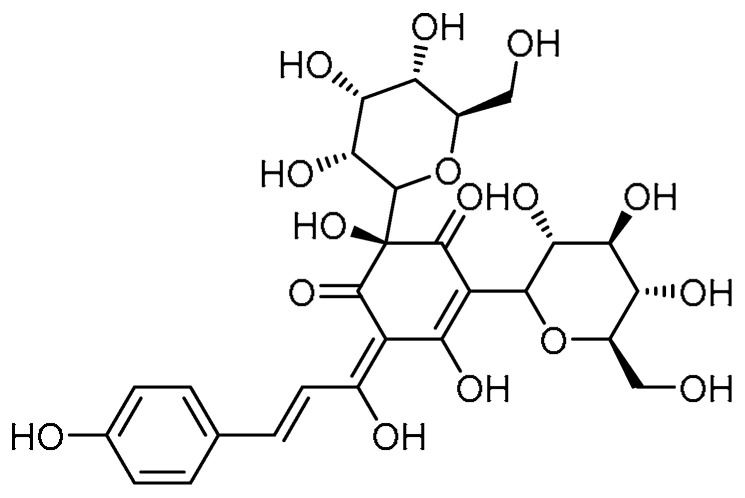	**17**	Hydroxysafflor yellow A	(2S,E)-2,5-Dihydroxy-6-((E)-1-hydroxy-3-(4-hydroxyphenyl)allylidene)-2,4-bis((3R,4R,5S,6R)-3,4,5-trihydroxy-6-(hydroxymethyl)tetrahydro-2H-pyran-2-yl)cyclohex-4-ene-1,3-dione	78281-02-4
18	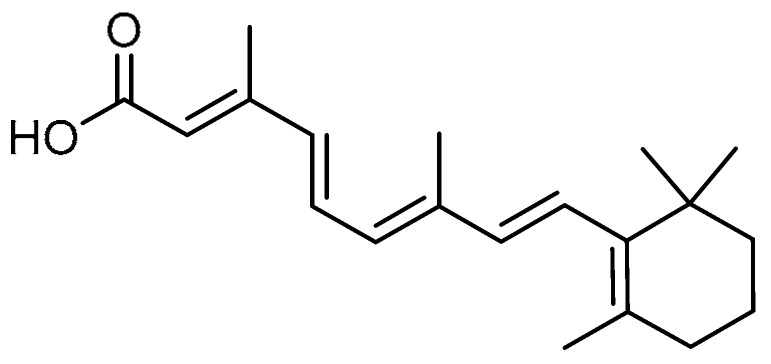	**18**	All-*trans* retinoic acid	(2E,4E,6E,8E)-3,7-Dimethyl-9-(2,6,6-trimethylcyclohex-1-en-1-yl)nona-2,4,6,8-tetraenoic acid	302-79-4
19	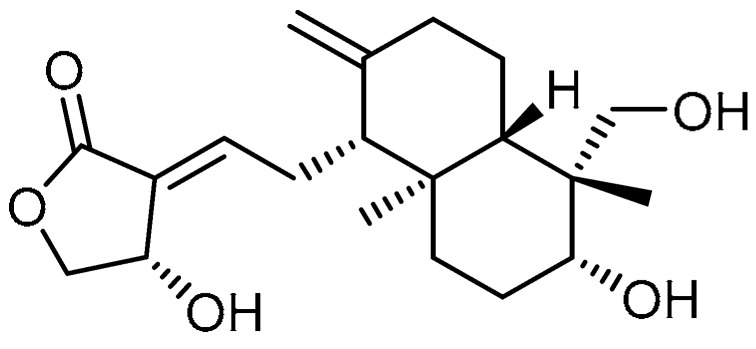	**19**	Andrographolide	3-[2-[Decahydro-6-hydroxy-5-(hydroxymethyl)-5,8a-dimethyl-2-methylene-1-napthalenyl]ethylidene]dihydro-4-hydroxy-2(3H)-furanone	5508-58-7
20	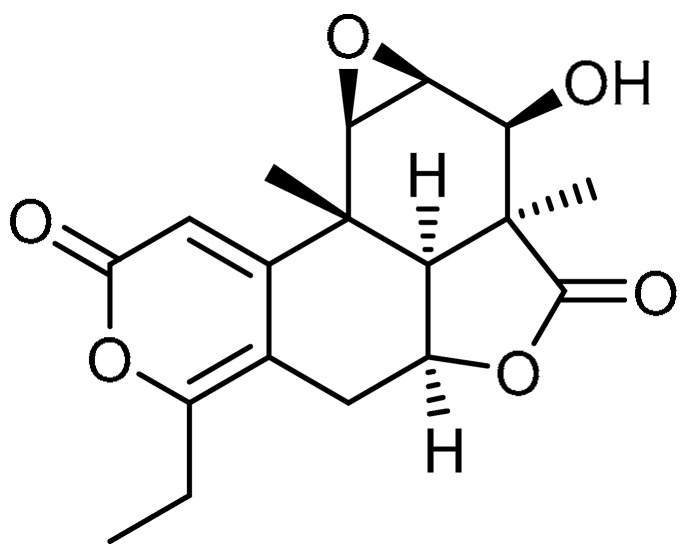	**20**	Nagilactone D	(1aS,2S,2aS,2a1S,4aS,9bR,9cR)-6-Ethyl-2-hydroxy-2a,9b-dimethyl-1a,2,2a,2a1,4a,5,9b,9c-octahydro-3H,8H-oxireno [2’,3’:5,6]isobenzofuro [7,1-fg]isochromene-3,8-dione	19891-53-3
21	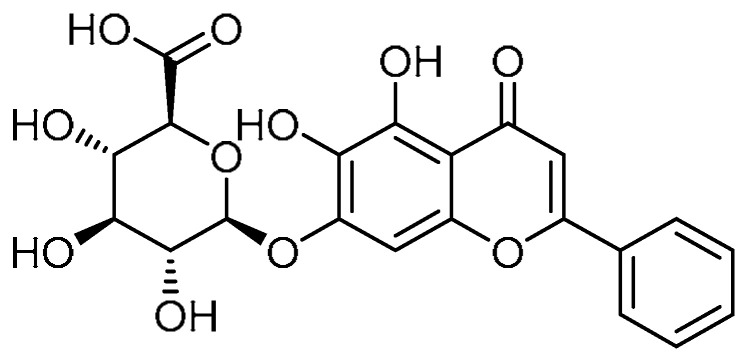	**21**	Baicalin	(2S,3S,4S,5R,6S)-6-[(5,6-Dihydroxy-4-oxo-2-phenyl-4H-1-benzopyran-7-yl)oxy]-3,4,5-trihydroxyoxane-2-carboxylic acid	21967-41-9
22	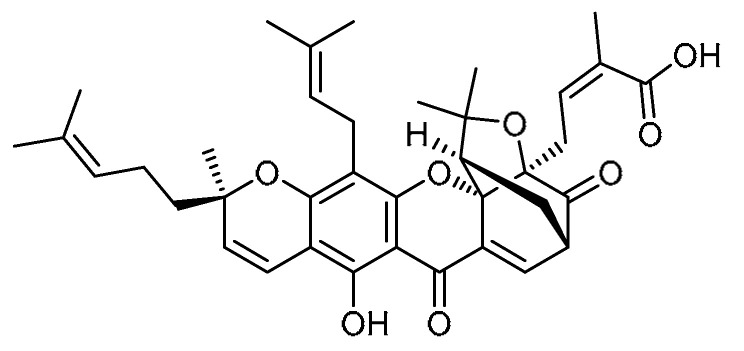	**22**	Gambogic acid	(2Z)-4-[(1R,3aS,5S,11R,14aS)-8-Hydroxy-3,3,11-trimethyl-13-(3-methylbut-2-en-1-yl)-11-(4-methylpent-3-en-1-yl)-7,15-dioxo-3a,4,5,7-tetrahydro-1H,3H,11H-1,5-methanofuro [3,4-g]pyrano [3,2-b]xanthen-1-yl]-2-methylbut-2-enoic acid	2752-65-0
23	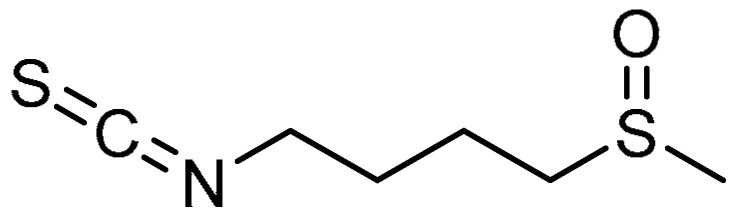	**23**	Sulforaphane	1-Isothiocyanato-4-(methanesulfinyl)butane	4478-93-7
24	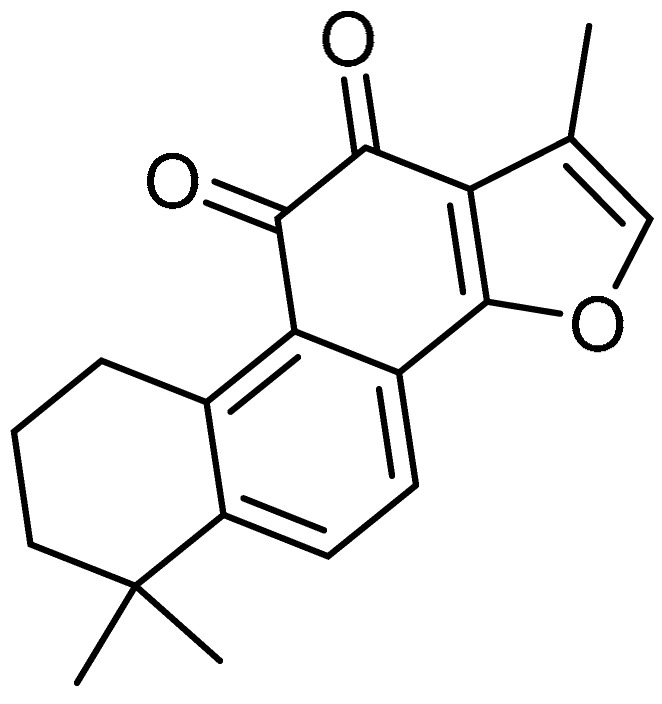	**24**	Tanshinone IIA	1,6,6-Trimethyl-6,7,8,9-tetrahydrophenanthro [1,2-b]furan-10,11-dione	568-72-9
25	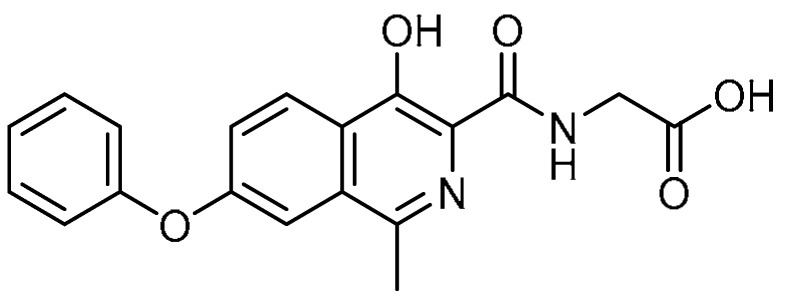	**25**	Roxadustat	2-[(4-Hydroxy-1-methyl-7-phenoxyisoquinoline-3-carbonyl)amino]acetic acid	808118-40-3
26	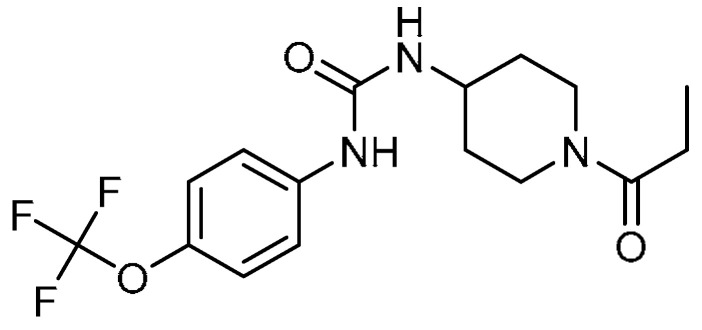	**26**	1-(1-Propanoylpiperidin-4-yl)-3-[4-(trifluoromethoxy)phenyl]urea	1-(1-Propionylpiperidin-4-yl)-3-(4-(trifluoromethoxy)phenyl)urea	1222780-33-7
27	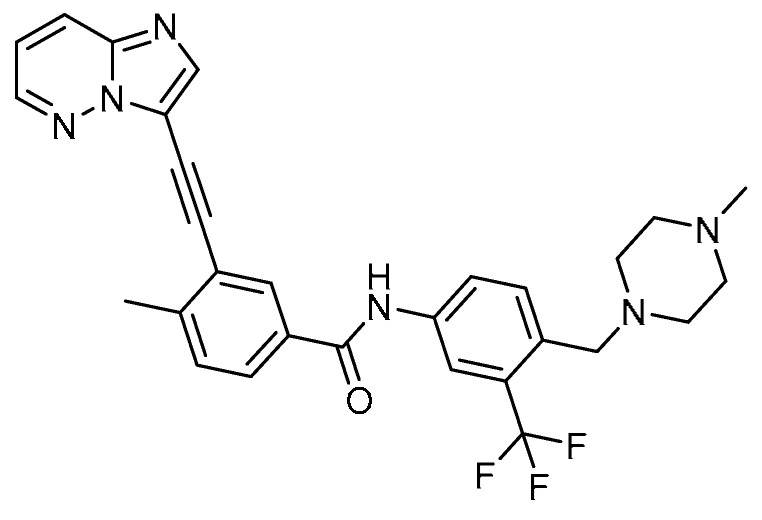	**27**	Ponatinib	3-(2-Imidazo [1,2-b]pyridazin-3-ylethynyl)-4-methyl-N-[4-[(4-methylpiperazin-1-yl)methyl]-3-(trifluoromethyl)phenyl]benzamide	943319-70-8
28	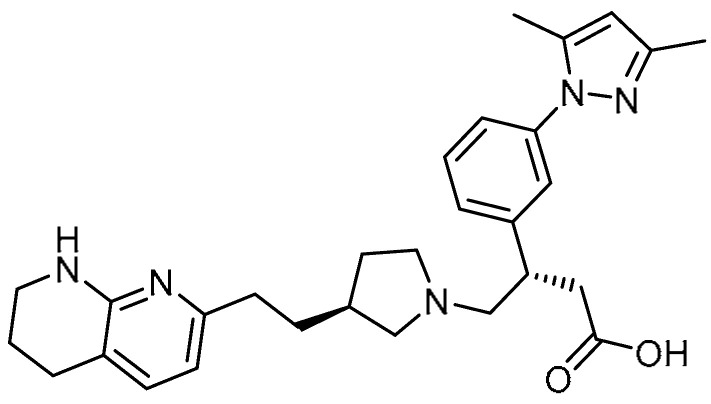	**28**	GSK3008348	(S)-3-(3-(3,5-Dimethyl-1H-pyrazol-1-yl)phenyl)-4-((S)-3-(2-(5,6,7,8-tetrahydro-1,8-naphthyridin-2-yl)ethyl)pyrrolidin-1-yl)butanoic acid	
29	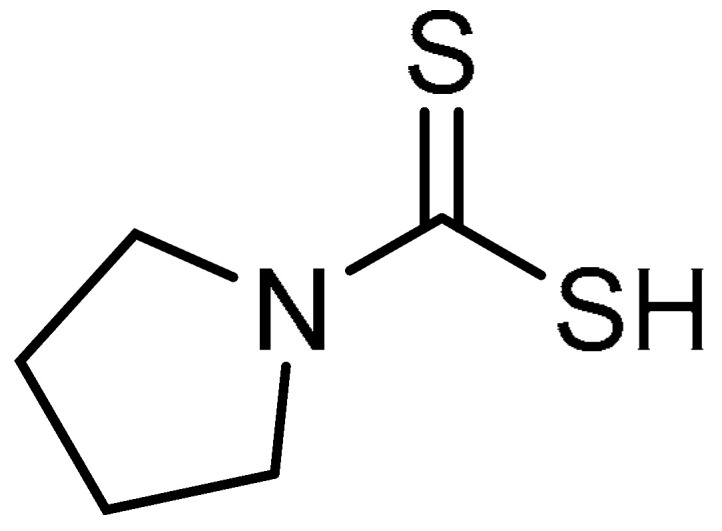	**29**	Pyrrolidine dithiocarbamate	Pyrrolidine-1-carbodithioic acid	25769-03-3
30	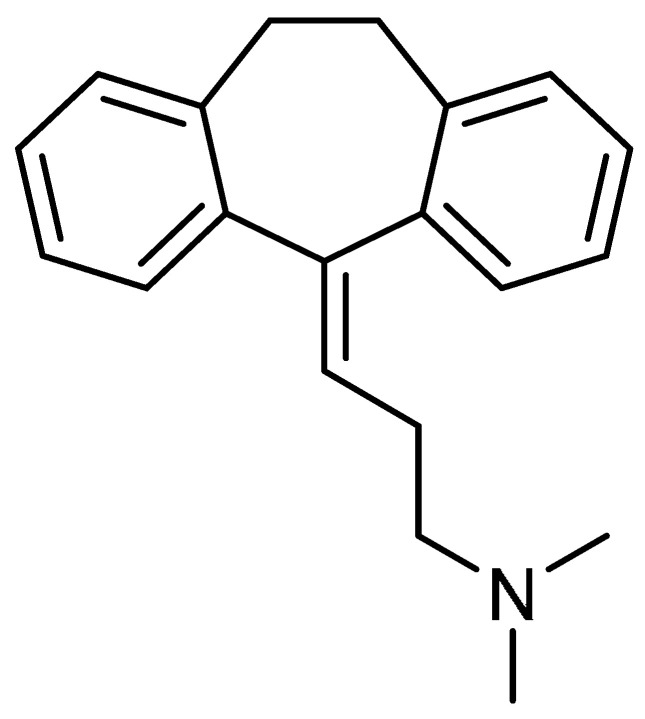	**30**	Amitriptyline	3-(10,11-Dihydro-5H-dibenzo[a,d]cycloheptene-5-ylidene)-N,N-dimethylpropan-1-amine	50-48-6
31	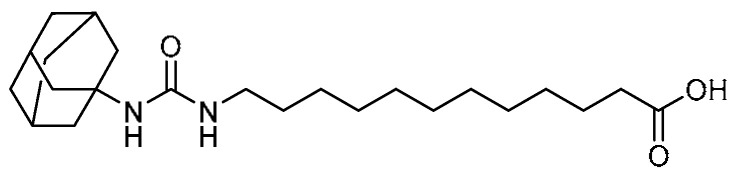	**31**	12-(3-(Adamantan-1-yl)ureido)dodecanoic acid	12-(3-(Adamantan-1-yl)ureido)dodecanoic acid	479413-70-2
32	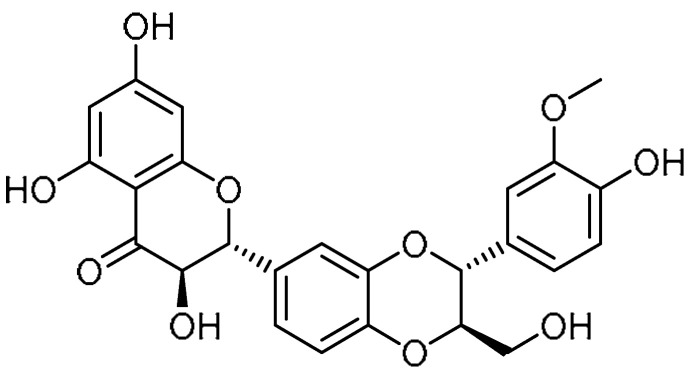	**32**	Silymarin	(2R,3R)-3,5,7-Trihydroxy-2-[(2R*,3R*)-3-(4-hydroxy-3-methoxyphenyl)-2-(hydroxymethyl)-2,3-dihydrobenzo[b][1,4]dioxin-6-yl]chroman-4-one	1265089-69-7
33	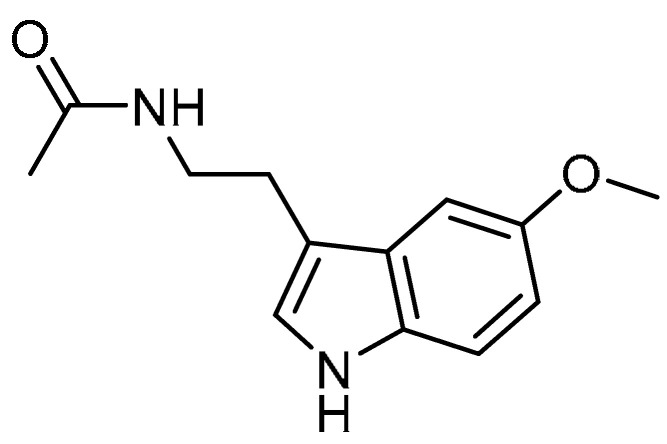	**33**	Melatonin	N-[2-(5-Methoxy-1H-indol-3-yl)ethyl]acetamide	73-31-4

## Data Availability

No new data were created or analyzed in this study. Data sharing is not applicable to this article.

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
