# Peer review of "Recent Advances (2015–2020) in Drug Discovery for Attenuation of Pulmonary Fibrosis and COPD"

_molecules, 2023, doi:10.3390/molecules28093674_

Round 1
Reviewer 1 Report
The manuscript presents a comprehensive review on drugs in the development for the treatment of pulmonary fibrosis and COPD. However, although the text is well writtent, the text and the references do not match. Please find below some examples where the text refers to subject and the reference has nothing to do with the mentioned subject in the text. This is really frustrating, to say the least, and make it really hard to evaluate the accuracy of the statements.
Recent studies witness that arsenic trioxide has been used as a drug for ulcer, malaria, and psoriasis [92]
[92] K. Krempaska, S. Barnowski, J. Gavini, N. Hobi, S. Ebener, C. Simillion, A. Stokes, R. Schliep, L. Knudsen, T.K. Geiser, M. Funke-Chambour, Azithromycin has enhanced effects on lung fibroblasts from idiopathic pulmonary fibrosis (IPF) patients compared to controls, Respiratory Research. 21 (2020) 25. https://doi.org/10.1186/s12931-020-1275-8.
The major cause of COPD is tobacco smoking and a few cases may be because of air pollution and genetics [98].
[98] G. Gartlehner, Efficacy and Safety of Inhaled Corticosteroids in Patients With COPD: A Systematic Review and Meta-Analysis of Health Outcomes, The Annals of Family Medicine. 4 (2006) 253–262. https://doi.org/10.1370/afm.517.
Approximately 2.4% of the global population are affected by COPD as of 2015 and in the year 2015 90% of deaths recorded in the developing countries [99].
[99] K.M. Kew, A. Seniukovich, Inhaled steroids and risk of pneumonia for chronic obstructive pulmonary disease, Cochrane Database of Systematic Reviews. (2014). https://doi.org/10.1002/14651858.CD010115.pub2.
When it comes treatment, only progress of the disease can be slowed down. However complete cure is still unknown [101,102].
[101] X.-S. Xie, M. Yang, H.-C. Liu, C. Zuo, H.-J. Li, J.-M. Fan, Ginsenoside Rg1, a major active component isolated from Panax notoginseng, restrains tubular epithelial to myofibroblast transition in vitro, Journal of Ethnopharmacology. 122 (2009) 35–41. https://doi.org/10.1016/j.jep.2008.11.020. 1219
[102] M. YU, X. YU, D. GUO, B. YU, L. LI, Q. LIAO, R. XING, Ginsenoside Rg1 attenuates invasion and migration by inhibiting transforming growth factor-β1-induced epithelial to mesenchymal transition in HepG2 cells, Molecular Medicine Reports. 11 (2015) 3167–3173. https://doi.org/10.3892/mmr.2014.3098. 1222
Amongst the know medications, corticosteroids are usually given to decrease acute exacerbations in patients with either moderate or severe disease [103].
[103] S. Guan, W. Xu, F. Han, W. Gu, L. Song, W. Ye, Q. Liu, X. Guo, Ginsenoside Rg1 Attenuates Cigarette Smoke-Induced Pulmonary Epithelial-Mesenchymal Transition via Inhibition of the TGF- β 1/Smad Pathway, BioMed Research International. 2017 (2017) 1–12. https://doi.org/10.1155/2017/7171404.
However, inhalation of steroids is associated with increased rates of pneumonia [104] and furthermore, long-term treatment with steroid tablets would result in significant side effects [105].
[104] C. Park, S.H. Hong, G.-Y. Kim, Y.H. Choi, So-Cheong-Ryong-Tang induces apoptosis through activation of the intrinsic and extrinsic apoptosis pathways, and inhibition of the PI3K/Akt signaling pathway in non-small-cell lung cancer A549 cells BMC Complementary and Alternative Medicine. 15 (2015) 113-. https://doi.org/10.1186/s12906-015-0639-y. 1228
[105] N.-R. Shin, C. Kim, C.-S. Seo, J.-W. Ko, Y.-K. Cho, J.-C. Kim, J.-S. Kim, I.-S. Shin, So-Cheong-Ryoung-Tang Attenuates Pulmonary Inflammation Induced by Cigarette Smoke in Bronchial Epithelial Cells and Experimental Mice, Frontiers in Pharmacology. 9 (2018) 1064-undefined. https://doi.org/10.3389/fphar.2018.01064.
Author Response
Dear Reviewer,
Thank you very much for the feedback on our paper titled “Recent advances (2015-20) in drug discovery for attenuation of pulmonary fibrosis and COPD”. We are highly grateful to you, editor and the fellow reviewers for taking time to review our manuscript and providing nice suggestions to further revise this manuscript. The necessary amendments are highlighted in the files named as revised manuscript with highlights. All the comments have been considered carefully and improvements have been made. We are uploading the separate reply letter to address all the reviewer's comments, please see the attachment.
We hope that you will find the revised version of the manuscript satisfactory for publication. Should there be any additional improvements necessary, please do not hesitate to contact us.
Thank you,
Regards,
Dr. Manikantha Maraswami.

Reviewer 2 Report
The review manuscript entitled “Recent advances (2015-20) in drug discovery for attenuation of pulmonary fibrosis and COPD” is an informative review work for the readers of the “molecules”. However, some points could be modified before its acceptance for publication.
1. There are more than 100 review papers published since 2015:
111 document results
TITLE ( pulmonary AND fibrosis AND treatment ) AND ( LIMIT-TO ( DOCTYPE , "re" ) ) AND ( LIMIT-TO ( PUBYEAR , 2023 ) OR LIMIT-TO ( PUBYEAR , 2022 ) OR LIMIT-TO ( PUBYEAR , 2021 ) OR LIMIT-TO ( PUBYEAR , 2020 ) OR LIMIT-TO ( PUBYEAR , 2019 ) OR LIMIT-TO ( PUBYEAR , 2018 ) OR LIMIT-TO ( PUBYEAR , 2017 ) OR LIMIT-TO ( PUBYEAR , 2016 ) OR LIMIT-TO ( PUBYEAR , 2015 ) )
(2023) Pneumologie, 77 (2), pp. 94-119.
https://www.scopus.com/inward/record.uri?eid=2-s2.0-85148114806&doi=10.1055%2fa-1983-6796&partnerID=40&md5=ebb1dfcf57920cda45436e9e249d3f23
(2023) Life, 13 (2), art. no. 486, .
https://www.scopus.com/inward/record.uri?eid=2-s2.0-85149256467&doi=10.3390%2flife13020486&partnerID=40&md5=493862c32243494fa32580778e7aa1eb
(2023) Pulmonary Therapy, .
https://www.scopus.com/inward/record.uri?eid=2-s2.0-85147776592&doi=10.1007%2fs41030-023-00216-0&partnerID=40&md5=568f63980406fa77555b60c608a32334
(2022) Processes, 10 (12), art. no. 2477, .
https://www.scopus.com/inward/record.uri?eid=2-s2.0-85144843197&doi=10.3390%2fpr10122477&partnerID=40&md5=c7f8eb8239a2acc0e942ed548165b9c4
(2022) Frontiers in Pharmacology, 13, art. no. 1059434, .
https://www.scopus.com/inward/record.uri?eid=2-s2.0-85143176395&doi=10.3389%2ffphar.2022.1059434&partnerID=40&md5=83bc1d8f089b16804b713fc61f302d20
(2022) Antioxidants, 11 (9), art. no. 1685, .
https://www.scopus.com/inward/record.uri?eid=2-s2.0-85138548178&doi=10.3390%2fantiox11091685&partnerID=40&md5=c6bc45ce490347e75788dab8f028db2a
(2022) Frontiers in Pharmacology, 13, art. no. 927653, .
https://www.scopus.com/inward/record.uri?eid=2-s2.0-85137993773&doi=10.3389%2ffphar.2022.927653&partnerID=40&md5=374a867a05136077584487d59a816b6d
(2022) Cells, 11 (16), art. no. 2489, .
https://www.scopus.com/inward/record.uri?eid=2-s2.0-85136603897&doi=10.3390%2fcells11162489&partnerID=40&md5=2301c97604e02237bac82b99c32c3703
(2022) JAMA, 328 (1), pp. 69-70.
https://www.scopus.com/inward/record.uri?eid=2-s2.0-85133264723&doi=10.1001%2fjama.2022.8492&partnerID=40&md5=8be46d045f6d2c4ba22813da6b8ae1b8
(2022) Cells, 11 (9), art. no. 1543, .
https://www.scopus.com/inward/record.uri?eid=2-s2.0-85129420013&doi=10.3390%2fcells11091543&partnerID=40&md5=1a1df850b6026fdfef4925480d1db76f
(2022) Surgery Today, 52 (5), pp. 736-744.
https://www.scopus.com/inward/record.uri?eid=2-s2.0-85111713595&doi=10.1007%2fs00595-021-02343-0&partnerID=40&md5=c82278b5502da6a2bc9f8402c0027a39
(2022) Biomedicines, 10 (4), art. no. 756, .
https://www.scopus.com/inward/record.uri?eid=2-s2.0-85127546766&doi=10.3390%2fbiomedicines10040756&partnerID=40&md5=e1be3eb3bafe80068833d3a37bce751d
(2022) PLoS ONE, 17 (3 March), art. no. e0265006, .
https://www.scopus.com/inward/record.uri?eid=2-s2.0-85125760462&doi=10.1371%2fjournal.pone.0265006&partnerID=40&md5=7107dedae0178986049562e838600e33
(2022) Clinical Respiratory Journal, 16 (2), pp. 84-96.
https://www.scopus.com/inward/record.uri?eid=2-s2.0-85122679448&doi=10.1111%2fcrj.13466&partnerID=40&md5=6677964ad26ad074f5cba576fb7ddd58
(2022) Russian Archives of Internal Medicine, 12 (4), pp. 267-275.
https://www.scopus.com/inward/record.uri?eid=2-s2.0-85136705755&doi=10.20514%2f2226-6704-2022-12-4-267-275&partnerID=40&md5=5a8d5d3b572b62e8316df68162f0c9d4
(2022) Expert Review of Respiratory Medicine, 16 (5), pp. 541-553.
https://www.scopus.com/inward/record.uri?eid=2-s2.0-85133275735&doi=10.1080%2f17476348.2022.2089116&partnerID=40&md5=4ac34c4f4c7ebddd2108f5eb9514cd2e
(2022) Thorax, art. no. 217976, .
https://www.scopus.com/inward/record.uri?eid=2-s2.0-85128960817&doi=10.1136%2fthoraxjnl-2021-217976&partnerID=40&md5=a1d03a75ca8c0fb7aaf28d79ce80429e
(2022) Oxidative Medicine and Cellular Longevity, 2022, art. no. 6197219, .
https://www.scopus.com/inward/record.uri?eid=2-s2.0-85127255756&doi=10.1155%2f2022%2f6197219&partnerID=40&md5=6e8b7af012c1eb06176fffc764669933
(2022) Acta Pharmaceutica Sinica B, 12 (1), pp. 18-32.
https://www.scopus.com/inward/record.uri?eid=2-s2.0-85114406557&doi=10.1016%2fj.apsb.2021.07.023&partnerID=40&md5=04f18fed891ae579ee2ceafee8d95b1e
(2021) BMJ Open, 11 (12), art. no. e050004, .
https://www.scopus.com/inward/record.uri?eid=2-s2.0-85122639035&doi=10.1136%2fbmjopen-2021-050004&partnerID=40&md5=3e5f40d37ea821d0ef5ed451a63a9fba
(2021) Frontiers in Pharmacology, 12, art. no. 707491, .
https://www.scopus.com/inward/record.uri?eid=2-s2.0-85114224758&doi=10.3389%2ffphar.2021.707491&partnerID=40&md5=904506b819627fd5ae299666576fdcc3
(2021) BMJ Open, 11 (8), art. no. e050578, .
https://www.scopus.com/inward/record.uri?eid=2-s2.0-85113174661&doi=10.1136%2fbmjopen-2021-050578&partnerID=40&md5=cce4261cd63cb5ceb240768c2061f25d
(2021) Antibiotics, 10 (5), art. no. 486, .
https://www.scopus.com/inward/record.uri?eid=2-s2.0-85105159418&doi=10.3390%2fantibiotics10050486&partnerID=40&md5=03ab92f6cff3db273df46360a6572769
(2021) American Journal of Respiratory and Critical Care Medicine, 203 (9), pp. 1065-1067.
https://www.scopus.com/inward/record.uri?eid=2-s2.0-85105047804&doi=10.1164%2frccm.202010-3994ED&partnerID=40&md5=c4ebfd59dd15b16b53957180a44a466b
(2021) Chinese Journal of Radiological Medicine and Protection, 41 (4), pp. 309-314.
https://www.scopus.com/inward/record.uri?eid=2-s2.0-85110382220&doi=10.3760%2fcma.j.issn.0254-5098.2021.04.013&partnerID=40&md5=533501b52e0f06d96431546deabf490e
(2021) Deutsches Arzteblatt International, 118 (9), pp. 152-162.
https://www.scopus.com/inward/record.uri?eid=2-s2.0-85110596497&doi=10.3238%2farztebl.m2021.0018&partnerID=40&md5=3677841b46549b9675cf0086090b99b2
(2021) Journal of Medical Virology, 93 (3), pp. 1378-1386.
https://www.scopus.com/inward/record.uri?eid=2-s2.0-85094653603&doi=10.1002%2fjmv.26634&partnerID=40&md5=65ac954886ac6ce0d7f512cb9ae6dd00
(2021) Food and Function, 12 (3), pp. 990-1007.
https://www.scopus.com/inward/record.uri?eid=2-s2.0-85100934701&doi=10.1039%2fd0fo03001e&partnerID=40&md5=6bf3e19d217c4edc35247ad5b26d0426
(2021) American Journal of Chinese Medicine, 49 (8), pp. 1965-1999.
https://www.scopus.com/inward/record.uri?eid=2-s2.0-85123037272&doi=10.1142%2fS0192415X21500932&partnerID=40&md5=06be38ac38af3f740e2b6b55b897573d
(2021) Current Drug Targets, 22 (7), pp. 793-802.
https://www.scopus.com/inward/record.uri?eid=2-s2.0-85107085692&doi=10.2174%2f1874609813666200928141822&partnerID=40&md5=db084a0dca3030389f7a928d62d0401d
(2021) Current Medicinal Chemistry, 28 (11), pp. 2234-2247.
https://www.scopus.com/inward/record.uri?eid=2-s2.0-85105113821&doi=10.2174%2f0929867327999200730173748&partnerID=40&md5=323900220d5b3fb8868a50153b44d5b9
(2021) European Respiratory Journal, 57 (1), art. no. 57, .
https://www.scopus.com/inward/record.uri?eid=2-s2.0-85099264988&doi=10.1183%2f13993003.03551-2020&partnerID=40&md5=c6331daa3a722bb146f640da8c5c2ac1
(2020) Frontiers in Pharmacology, 11, art. no. 607689, .
https://www.scopus.com/inward/record.uri?eid=2-s2.0-85098519478&doi=10.3389%2ffphar.2020.607689&partnerID=40&md5=2d004e1ebd6861bd452e2fe1c9714840
(2020) Current opinion in pulmonary medicine, 26 (6), pp. 679-684.
https://www.scopus.com/inward/record.uri?eid=2-s2.0-85092750590&doi=10.1097%2fMCP.0000000000000730&partnerID=40&md5=02c44dd9cf32b6a42cd81f2a9072d505
(2020) Chinese Journal of New Drugs, 29 (20), pp. 2389-2394.
https://www.scopus.com/inward/record.uri?eid=2-s2.0-85097824142&partnerID=40&md5=1543bfe9bc69d36c5aa9f52dface95de
(2020) Respiratory Investigation, 58 (5), pp. 320-335.
https://www.scopus.com/inward/record.uri?eid=2-s2.0-85085596343&doi=10.1016%2fj.resinv.2020.04.002&partnerID=40&md5=3db205d241b74f04753e3a9e1aad46c6
(2020) Medicine (United States), 99 (31), p. E21310.
https://www.scopus.com/inward/record.uri?eid=2-s2.0-85089171391&doi=10.1097%2fMD.0000000000021310&partnerID=40&md5=6c6d2f9cea3d17befb35bfb885384854
(2020) Current opinion in pulmonary medicine, 26 (4), pp. 363-371.
https://www.scopus.com/inward/record.uri?eid=2-s2.0-85086481723&doi=10.1097%2fMCP.0000000000000684&partnerID=40&md5=2631c67192d02bac8cedc0e4c07d1098
(2020) Journal of Clinical Medicine, 9 (6), art. no. 1917, pp. 1-20.
https://www.scopus.com/inward/record.uri?eid=2-s2.0-85088393957&doi=10.3390%2fjcm9061917&partnerID=40&md5=8233b8f7efa34329b10750c717fcbfb0
(2020) Pneumologe, 17 (3), pp. 186-196.
https://www.scopus.com/inward/record.uri?eid=2-s2.0-85083393985&doi=10.1007%2fs10405-020-00313-w&partnerID=40&md5=ff1fe175b48efba94122481ed4679555
(2020) Pneumologe, 17 (3), pp. 177-185.
https://www.scopus.com/inward/record.uri?eid=2-s2.0-85078177194&doi=10.1007%2fs10405-019-00296-3&partnerID=40&md5=984fa4141c3c5b977a7444e502c13098
(2020) Cochrane Database of Systematic Reviews, 2020 (4), art. no. CD009529, .
https://www.scopus.com/inward/record.uri?eid=2-s2.0-85082979111&doi=10.1002%2f14651858.CD009529.pub4&partnerID=40&md5=9821f8357eb2bdc877dde19d7a793d26
(2020) BMC Pulmonary Medicine, 20 (1), art. no. 57, .
https://www.scopus.com/inward/record.uri?eid=2-s2.0-85080997607&doi=10.1186%2fs12890-020-1092-3&partnerID=40&md5=5a8b7fb9b5a7320f55872a7f364ea075
(2020) Chinese Journal of Laboratory Medicine, 43 (1), pp. 91-95.
https://www.scopus.com/inward/record.uri?eid=2-s2.0-85085335395&doi=10.3760%2fcma.j.issn.1009-9158.2020.01.011&partnerID=40&md5=93369c85abc5c5d35ff3df127f09b27c
(2020) Advances in Respiratory Medicine, 88 (6), pp. 599-607.
https://www.scopus.com/inward/record.uri?eid=2-s2.0-85099267878&doi=10.5603%2fARM.A2020.0190&partnerID=40&md5=37e49a2ff7b3edb5231d28e5ba12fbea
(2020) Frontiers in Pharmacology, 11, art. no. 415, .
https://www.scopus.com/inward/record.uri?eid=2-s2.0-85084816518&doi=10.3389%2ffphar.2020.00415&partnerID=40&md5=c1d917c121e9df39c8f3b34aaa20daea
(2020) Drug, Healthcare and Patient Safety, 12, pp. 85-94.
https://www.scopus.com/inward/record.uri?eid=2-s2.0-85084355345&doi=10.2147%2fDHPS.S224007&partnerID=40&md5=ba4265f4649eee10e1b0beac3c50e04e
(2020) Pediatric Pulmonology, 55 (1), pp. 33-57.
https://www.scopus.com/inward/record.uri?eid=2-s2.0-85074422688&doi=10.1002%2fppul.24537&partnerID=40&md5=ca2a6e3fa56b47016eba1dc66a261ecb
(2019) Frontiers in Bioengineering and Biotechnology, 7, art. no. 406, .
https://www.scopus.com/inward/record.uri?eid=2-s2.0-85077325961&doi=10.3389%2ffbioe.2019.00406&partnerID=40&md5=c4e134ceb5203ab11ac83d3e52535e58
(2019) Archivos de Bronconeumologia, 55 (12), pp. 642-647.
https://www.scopus.com/inward/record.uri?eid=2-s2.0-85067615111&doi=10.1016%2fj.arbres.2019.05.008&partnerID=40&md5=9f9ba4e58f748145f5e7c8e405612b34
(2019) Journal of Clinical Medicine, 8 (10), art. no. 1547, .
https://www.scopus.com/inward/record.uri?eid=2-s2.0-85090603437&doi=10.3390%2fjcm8101547&partnerID=40&md5=f9c5c37b28db9b8b3a9ed70b61b61d3c
(2019) The American journal of managed care, 25 (11), pp. S195-S203.
https://www.scopus.com/inward/record.uri?eid=2-s2.0-85071580555&partnerID=40&md5=25716874a2f4924db5bd535c05c165c3
(2019) Medicine (United States), 98 (30), art. no. e16325, .
https://www.scopus.com/inward/record.uri?eid=2-s2.0-85070580718&doi=10.1097%2fMD.0000000000016325&partnerID=40&md5=f8016bf93b58d3c1045e2b5133c5cd07
(2019) Medicine (United States), 98 (17), art. no. e15407, .
https://www.scopus.com/inward/record.uri?eid=2-s2.0-85065322970&doi=10.1097%2fMD.0000000000015407&partnerID=40&md5=fe895b7701da11557305a337364dec28
(2019) Expert Review of Respiratory Medicine, 13 (3), pp. 229-239.
https://www.scopus.com/inward/record.uri?eid=2-s2.0-85061513931&doi=10.1080%2f17476348.2019.1568244&partnerID=40&md5=02419cb7b8647a8988b754eefb86b61a
(2019) Chinese Journal of New Drugs, 28 (2), pp. 238-243.
https://www.scopus.com/inward/record.uri?eid=2-s2.0-85065088774&partnerID=40&md5=bf530ace708b471af9eff36cdbfa56a2
(2019) Journal of Thoracic Disease, 11, pp. S1740-S1754.
https://www.scopus.com/inward/record.uri?eid=2-s2.0-85073798014&doi=10.21037%2fjtd.2019.04.62&partnerID=40&md5=a1b1f4a82880a172b884083054ad6895
(2019) Evidence-based Complementary and Alternative Medicine, 2019, art. no. 5170638, .
https://www.scopus.com/inward/record.uri?eid=2-s2.0-85068155450&doi=10.1155%2f2019%2f5170638&partnerID=40&md5=4585113dd553b3cc060eb2ed40153a54
(2019) Therapeutics and Clinical Risk Management, 15, pp. 73-81.
https://www.scopus.com/inward/record.uri?eid=2-s2.0-85060583663&doi=10.2147%2fTCRM.S160248&partnerID=40&md5=a5bb9a835fd0383bc64485952473cbee
(2019) Translational Oncology, 12 (1), pp. 162-169.
https://www.scopus.com/inward/record.uri?eid=2-s2.0-85054918929&doi=10.1016%2fj.tranon.2018.09.009&partnerID=40&md5=e90771224ec679d51e138a61b3137fe0
(2018) Medicine (United States), 97 (44), art. no. e13077, .
https://www.scopus.com/inward/record.uri?eid=2-s2.0-85055906981&doi=10.1097%2fMD.0000000000013077&partnerID=40&md5=2cec4845313aff79f113bc0117fea60b
(2018) Pharmaceutical Care and Research, 18 (4), pp. 241-246.
https://www.scopus.com/inward/record.uri?eid=2-s2.0-85055872009&doi=10.5428%2fpcar20180401&partnerID=40&md5=4dcacdddb1d703e6d8c2a6cca8202f92
(2018) Cochrane Database of Systematic Reviews, 2018 (7), art. no. CD011581, .
https://www.scopus.com/inward/record.uri?eid=2-s2.0-85050677994&doi=10.1002%2f14651858.CD011581.pub3&partnerID=40&md5=6749f1d73fb6b4760bc94068ce04a9c5
(2018) Medical sciences (Basel, Switzerland), 6 (3), .
https://www.scopus.com/inward/record.uri?eid=2-s2.0-85149160850&doi=10.3390%2fmedsci6030059&partnerID=40&md5=b9eb5b0699628e5a4c128ccd2195b018
(2018) Respiratory Investigation, 56 (4), pp. 268-291.
https://www.scopus.com/inward/record.uri?eid=2-s2.0-85049301528&doi=10.1016%2fj.resinv.2018.03.003&partnerID=40&md5=14986679bd2590159941e8d5d5aa4501
(2018) JCI insight, 3 (10), .
https://www.scopus.com/inward/record.uri?eid=2-s2.0-85062249430&doi=10.1172%2fjci.insight.120362&partnerID=40&md5=fe50cb5e5f80251e2819c0f4ec836b06
(2018) Frontiers in Medicine, 5 (MAY), art. no. 142, .
https://www.scopus.com/inward/record.uri?eid=2-s2.0-85050117447&doi=10.3389%2ffmed.2018.00142&partnerID=40&md5=9b1cbec97ca13c85edc9d3976a6d865f
(2018) Expert Opinion on Emerging Drugs, 23 (2), pp. 159-172.
https://www.scopus.com/inward/record.uri?eid=2-s2.0-85048461896&doi=10.1080%2f14728214.2018.1471465&partnerID=40&md5=980cb8de5fe587c0a9bd0134247aef4a
(2018) Drugs in R and D, 18 (1), pp. 19-25.
https://www.scopus.com/inward/record.uri?eid=2-s2.0-85037632870&doi=10.1007%2fs40268-017-0221-9&partnerID=40&md5=167c04c7ee314e1d04c2e3a7c1bdd579
(2018) European Respiratory Journal, 52 (2), art. no. 1801287, .
https://www.scopus.com/inward/record.uri?eid=2-s2.0-85062617760&doi=10.1183%2f13993003.01287-2018&partnerID=40&md5=7b2dfb17c73ad864b4575dbce9162b7a
(2017) Praxis, 106 (18), pp. 999-1006.
https://www.scopus.com/inward/record.uri?eid=2-s2.0-85028809094&doi=10.1024%2f1661-8157%2fa002770&partnerID=40&md5=309394a0bdc8d37320699d7474dad31a
(2017) Biomedicine and Pharmacotherapy, 93, pp. 666-673.
https://www.scopus.com/inward/record.uri?eid=2-s2.0-85021707622&doi=10.1016%2fj.biopha.2017.06.052&partnerID=40&md5=8fdcf26ce5855f265a47f4e4e8aa1bbf
(2017) Current Opinion in Pulmonary Medicine, 23 (5), pp. 418-425.
https://www.scopus.com/inward/record.uri?eid=2-s2.0-85020488964&doi=10.1097%2fMCP.0000000000000408&partnerID=40&md5=0b7b849460f309f112ee2e636d423dad
(2017) Southern Medical Journal, 110 (6), pp. 393-398.
https://www.scopus.com/inward/record.uri?eid=2-s2.0-85021859455&doi=10.14423%2fSMJ.0000000000000655&partnerID=40&md5=a8d4095b509fed5181b8b8117163efbf
(2017) Expert Review of Clinical Pharmacology, 10 (5), pp. 483-491.
https://www.scopus.com/inward/record.uri?eid=2-s2.0-85014567984&doi=10.1080%2f17512433.2017.1295846&partnerID=40&md5=a6d352424e6fccac16628a7aecdc2c1c
(2017) Therapeutic Advances in Respiratory Disease, 11 (5), pp. 193-209.
https://www.scopus.com/inward/record.uri?eid=2-s2.0-85019028473&doi=10.1177%2f1753465817691239&partnerID=40&md5=d22598443ee222f321982123a84bb75f
(2017) Respiratory Medicine, 123, pp. 98-104.
https://www.scopus.com/inward/record.uri?eid=2-s2.0-85007417163&doi=10.1016%2fj.rmed.2016.12.016&partnerID=40&md5=7fd90ed565bd269c447fb201ca1c66af
(2017) Journal of Internal Medicine, 281 (2), pp. 149-166.
https://www.scopus.com/inward/record.uri?eid=2-s2.0-85003946087&doi=10.1111%2fjoim.12571&partnerID=40&md5=44682139c43241e7a2d3e1aca667ef19
(2017) Journal of Clinical Medicine, 6 (1), .
https://www.scopus.com/inward/record.uri?eid=2-s2.0-85067220110&doi=10.3390%2fjcm6010002&partnerID=40&md5=5ac1053053ec541d7c916c7f90ad5af2
(2017) Clinical Medicine Insights: Therapeutics, 9, .
https://www.scopus.com/inward/record.uri?eid=2-s2.0-85045020247&doi=10.1177%2f1179559X17719126&partnerID=40&md5=678bb2cd511a6c0e5813d10469643bf7
(2017) Studia Pneumologica et Phthiseologica, 77 (5), pp. 190-193.
https://www.scopus.com/inward/record.uri?eid=2-s2.0-85039844600&partnerID=40&md5=42b9b47cb47054fadd84829e0fd100fc
(2017) Frontiers in Medicine, 4 (SEP), art. no. 154, .
https://www.scopus.com/inward/record.uri?eid=2-s2.0-85038447908&doi=10.3389%2ffmed.2017.00154&partnerID=40&md5=44f8e83acdeb31f4f704626d42c9cc15
(2016) Expert Review of Respiratory Medicine, 10 (11), pp. 1221-1228.
https://www.scopus.com/inward/record.uri?eid=2-s2.0-84992166446&doi=10.1080%2f17476348.2017.1246963&partnerID=40&md5=2bcb3c1ec8fdae7f5df84428a86662f3
(2016) Expert Opinion on Drug Safety, 15 (11), pp. 1483-1489.
https://www.scopus.com/inward/record.uri?eid=2-s2.0-84992125528&doi=10.1080%2f14740338.2016.1218470&partnerID=40&md5=98a73f0d952e88fe4d1a49d9c9859e41
(2016) Journal of Clinical Medicine, 5 (9), art. no. 78, .
https://www.scopus.com/inward/record.uri?eid=2-s2.0-85014767593&doi=10.3390%2fjcm5090078&partnerID=40&md5=76d2946196bbbe95f1a930e41fb8fb96
(2016) Paediatric Respiratory Reviews, 20, pp. 6-7.
https://www.scopus.com/inward/record.uri?eid=2-s2.0-84977482360&doi=10.1016%2fj.prrv.2016.06.004&partnerID=40&md5=645f5b4ef9a8a7aae914414a62619d67
(2016) Core Evidence, 11, pp. 11-22.
https://www.scopus.com/inward/record.uri?eid=2-s2.0-84991648540&doi=10.2147%2fCE.S76549&partnerID=40&md5=b09a19e8f1c1fe8ad89796278b2f19d3
(2016) European Respiratory Journal, 47 (5), pp. 1321-1323.
https://www.scopus.com/inward/record.uri?eid=2-s2.0-84969786967&doi=10.1183%2f13993003.00389-2016&partnerID=40&md5=f92de2c4eb8cbacfb062b4cfe0946aa9
(2016) Therapeutics and Clinical Risk Management, 12, pp. 563-574.
https://www.scopus.com/inward/record.uri?eid=2-s2.0-84963827872&doi=10.2147%2fTCRM.S81144&partnerID=40&md5=1f794989220986d3612249d23057b912
(2016) Cochrane Database of Systematic Reviews, 2016 (3), art. no. CD011581, .
https://www.scopus.com/inward/record.uri?eid=2-s2.0-84961960660&doi=10.1002%2f14651858.CD011581.pub2&partnerID=40&md5=5f4c81d1ab66704c965c63b99804895b
(2016) Expert Opinion on Biological Therapy, 16 (3), pp. 375-387.
https://www.scopus.com/inward/record.uri?eid=2-s2.0-84957964217&doi=10.1517%2f14712598.2016.1124085&partnerID=40&md5=78c1ac96baaa76325ff9565b52eface6
(2016) Cochrane Database of Systematic Reviews, 2016 (1), art. no. CD009529, .
https://www.scopus.com/inward/record.uri?eid=2-s2.0-84970046053&doi=10.1002%2f14651858.CD009529.pub3&partnerID=40&md5=a3483879b0c88e8bec03399089363c77
(2016) JK Practitioner, 21 (1-2), pp. 1-3.
https://www.scopus.com/inward/record.uri?eid=2-s2.0-85071336779&partnerID=40&md5=1b224cad8d273c9c7aba81112db98b73
(2016) Pulmonologiya, 26 (4), pp. 399-419.
https://www.scopus.com/inward/record.uri?eid=2-s2.0-85046054928&doi=10.18093%2f0869-0189-2016-26-4-399-419&partnerID=40&md5=7d9401349b82ed3acaa2a9f1061912a6
(2016) Pneumologia, 65 (3), pp. 127-132.
https://www.scopus.com/inward/record.uri?eid=2-s2.0-84997270847&partnerID=40&md5=29b4c91ca7008afb16edb78bc7332873
(2015) Drug Design, Development and Therapy, 9, pp. 6407-6419.
https://www.scopus.com/inward/record.uri?eid=2-s2.0-84949958099&doi=10.2147%2fDDDT.S76648&partnerID=40&md5=f045e8b20cb9b4b5c807b11778f60629
(2015) Pulmonary Therapy, 1 (1), pp. 19-30.
https://www.scopus.com/inward/record.uri?eid=2-s2.0-84979647809&doi=10.1007%2fs41030-015-0007-6&partnerID=40&md5=af8856d8e71482823656bfce72e292ba
(2015) Expert Opinion on Emerging Drugs, 20 (4), pp. 537-552.
https://www.scopus.com/inward/record.uri?eid=2-s2.0-84951908863&doi=10.1517%2f14728214.2015.1102886&partnerID=40&md5=da5c983203b016d7ee00bb90674bec47
(2015) Current Stem Cell Research and Therapy, 10 (6), pp. 466-476.
https://www.scopus.com/inward/record.uri?eid=2-s2.0-84947726077&doi=10.2174%2f1574888X10666150519092639&partnerID=40&md5=f89a54328299c40b9704cc7eebd0083c
(2015) Jornal Brasileiro de Pneumologia, 41 (5), pp. 454-466.
https://www.scopus.com/inward/record.uri?eid=2-s2.0-84947226663&doi=10.1590%2fS1806-37132015000000152&partnerID=40&md5=a0a72e35026f091baaa27b63c24f0910
(2015) Current Opinion in Pulmonary Medicine, 21 (5), pp. 479-489.
https://www.scopus.com/inward/record.uri?eid=2-s2.0-84942905255&doi=10.1097%2fMCP.0000000000000190&partnerID=40&md5=a8ba88802f1a8ca511f7c12341bb4757
(2015) Drug Discovery Today, 20 (5), pp. 514-524.
https://www.scopus.com/inward/record.uri?eid=2-s2.0-84939982550&doi=10.1016%2fj.drudis.2015.01.001&partnerID=40&md5=2c76dc16c5ca8368d552f3c6026b202b
(2015) European Respiratory Journal, 45 (5), pp. 1434-1445.
https://www.scopus.com/inward/record.uri?eid=2-s2.0-84928995543&doi=10.1183%2f09031936.00174914&partnerID=40&md5=23879f9e210b1c2780d6270641688866
(2015) Respiratory Investigation, 53 (3), pp. 88-92.
https://www.scopus.com/inward/record.uri?eid=2-s2.0-84928770758&doi=10.1016%2fj.resinv.2014.12.005&partnerID=40&md5=b518a0111340ad9366982b45835469c8
(2015) Stem Cells and Cloning: Advances and Applications, 8, pp. 61-65.
https://www.scopus.com/inward/record.uri?eid=2-s2.0-84930941044&doi=10.2147%2fSCCAA.S49801&partnerID=40&md5=29569b577102a514fcbbb218c5c447e5
(2015) Respiration, 89 (3), pp. 201-207.
https://www.scopus.com/inward/record.uri?eid=2-s2.0-84925256449&doi=10.1159%2f000369828&partnerID=40&md5=bb7457a286319aaf531626eb50d738ed
(2015) Clinical Medicine Insights: Circulatory, Respiratory and Pulmonary Medicine, 9s1, pp. 179-185.
https://www.scopus.com/inward/record.uri?eid=2-s2.0-85025066093&doi=10.4137%2fCCRPM.S23321&partnerID=40&md5=c4de54c4b599d29bfcdfa796e5ed3983
(2015) BioMed Research International, 2015, art. no. 329481, .
https://www.scopus.com/inward/record.uri?eid=2-s2.0-84953264465&doi=10.1155%2f2015%2f329481&partnerID=40&md5=917298cadf731af9b6573c14a7bb2f3a
(2015) Therapeutic Advances in Respiratory Disease, 9 (3), pp. 121-129.
https://www.scopus.com/inward/record.uri?eid=2-s2.0-84930694331&doi=10.1177%2f1753465815579365&partnerID=40&md5=dd6797b3f812b03f38c3c06516ceaae5
(2015) European Respiratory Review, 24 (135), pp. 65-68.
https://www.scopus.com/inward/record.uri?eid=2-s2.0-84923920911&doi=10.1183%2f09059180.00011414&partnerID=40&md5=888316d4ea84b90eea29869495c35340
(2015) European Respiratory Review, 24 (135), pp. 58-64.
https://www.scopus.com/inward/record.uri?eid=2-s2.0-84923876116&doi=10.1183%2f09059180.00011514&partnerID=40&md5=d11e2471b7f6b0d1b4cbc176695e1d3d
In introduction section, the readers expect to read some points dealing with recent reviews published in the literature, the gaps in the works and the focus of the present work. I would like to ask the authors to add a paragraph(s) dealing with this point.
2. Please define all abbreviations used in the manuscript in their 1st appearance and provide a full alphabetic ordered list of abbreviations.
3. Please use abbreviations for repeatedly used long phrases, such as pulmonary fibrosis (PF), idiopathic pulmonary fibrosis (IPF) etc.
4. Please collect all chemical structures in a Table rather than different figures, and list some more information of the drugs, such as: their used numbers in the text, name, structure, CAS number etc.
5. Please add some illustrative graphs dealing with various pathways involved in PF discussed in your work.
6. You reviewed the increase/decrease of some biomarkers in your manuscript. To provide a clear picture, in a separate sub-section, please provide a list of biomarkers and their changes and also measurement methods briefly. Possible tabulated summary could also be a nice presentation.
7. Some minor points have been marked on pdf file. Please consider them in your revision.

Author Response

(The authors gave the same response as above.)

Round 2
Reviewer 1 Report
The authors have corrected the references. My opinion is that the manuscript should be restricted to pulmonary fibrosis. The review on COPD is quite superficial as compared to that on pulmonary fibrosis.
Reviewer 2 Report
Revision looks fine!